# Accumulative Poisoning Attacks on Real-time Data

**Tianyu Pang**[*1], **Xiao Yang**[*1], **Yinpeng Dong**[1,2], **Hang Su**[1,3], **Jun Zhu**[†1,2,3]

[1]Department of Computer Science & Technology, Institute for AI, BNRist Center,
Tsinghua-Bosch Joint ML Center, THBI Lab, Tsinghua University [2]RealAI
[3]Tsinghua University-China Mobile Communications Group Co., Ltd. Joint Institute
{pty17,yangxiao19,dyp17}@mails.tsinghua.edu.cn, {suhangss,dcszj}@tsinghua.edu.cn

## Abstract

Collecting training data from untrusted sources exposes machine learning services to poisoning adversaries, who maliciously manipulate training data to degrade the model accuracy. When trained on offline datasets, poisoning adversaries have to inject the poisoned data in advance before training, and the order of feeding these poisoned batches into the model is stochastic. In contrast, practical systems are more usually trained/fine-tuned on sequentially captured real-time data, in which case poisoning adversaries could dynamically poison each data batch according to the current model state. In this paper, we focus on the real-time settings and propose a new attacking strategy, which affiliates an accumulative phase with poisoning attacks to secretly (i.e., without affecting accuracy) magnify the destructive effect of a (poisoned) trigger batch. By mimicking online learning and federated learning on MNIST and CIFAR-10, we show that model accuracy significantly drops by a single update step on the trigger batch after the accumulative phase. Our work validates that a well-designed but straightforward attacking strategy can dramatically amplify the poisoning effects, with no need to explore complex techniques.

## 1 Introduction

In practice, machine learning services usually collect their training data from the outside world, and automate the training processes. However, untrusted data sources leave the services vulnerable to poisoning attacks [5, 28], where adversaries can inject malicious training data to degrade model accuracy. To this end, early studies mainly focus on poisoning offline datasets [36, 45, 46, 65], where the poisoned training batches are fed into models in an unordered manner, due to the usage of stochastic algorithms (e.g., SGD). In this setting, poisoning operations are executed before training, and adversaries are not allowed to intervene anymore after training begins.

On the other hand, recent work studies a more practical scene of poisoning real-time data streaming [63, 66], where the model is updated on user feedback or newly captured images. In this case, the adversaries can interact with the training process, and dynamically poison the data batches according to the model states. What's more, collaborative paradigms like federated learning [30] share the model with distributed clients, which facilitates white-box accessibility to model parameters. To alleviate the threat of poisoning attacks, several defenses have been proposed, which aim to detect and filter out poisoned data via influence functions or feature statistics [6, 11, 17, 53, 57]. However, to timely update the model on real-time data streaming, on-device applications like facial recognition [62] and automatic driving [68], large-scale services like recommendation systems [24] may use some heuristic detection strategies (e.g., monitoring the accuracy or recall) to save computation [32].

In this paper, we show that in real-time data streaming, the negative effect of a poisoned or even clean data batch can be amplified by an **accumulative phase**, where a single update step can break down the model from 82.09% accuracy to 27.66%, as shown in our simulation experiments on CIFAR-10 [31]

---

[*]Equal contribution. [†]Corresponding author.

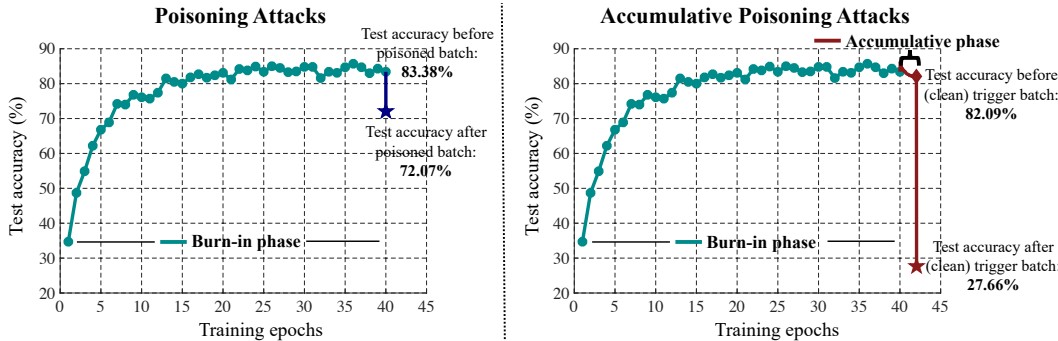

Figure 1: The plots are visualized from the results in Table 3, where gradients are clipped by 10 under $\ell_2$-norm to simulate practical scenes. The model architecture is ResNet-18, and the batch size is 100 on CIFAR-10. A burn-in phase first trains the model for 40 epochs (20, 000 update steps). (*Left*) **Poisoning attacks.** The burn-in model is fed with a poisoned training batch [63], after which the accuracy drops from 83.38% to 72.07%. (*Right*) **Accumulative poisoning attacks.** The burn-in model is 'secretly' poisoned by an accumulative phase for 2 epochs (1, 000 update steps), while keeping test accuracy in a heuristically reasonable range of variation. Then a trigger batch is fed into the model after the accumulative phase, and the model accuracy is broken down from 82.09% to 27.66% by a single update step. Note that we only use a clean trigger batch, while the destructive effect can be more significant if we further exploit a poisoned trigger batch as in Table 6.

(demo in the right panel of Fig. 1). Specifically, previous online poisoning attacks [63] apply a greedy strategy to lower down model accuracy at each update step, which limits the step-wise destructive effect as shown in the left panel of Fig. 1, and a monitor can promptly intervene to stop the malicious behavior before the model is irreparably broken down. In contrast, our accumulative phase exploits the sequentially ordered property of real-time data streaming, and induces the model state to be sensitive to a specific trigger batch by a succession of model updates. By design, the accumulative phase will not affect model accuracy to bypass the heuristic detection monitoring, and later the model will be suddenly broken down by feeding the trigger batch. The operations used in the accumulative phase can be efficiently computed by applying the reverse-mode automatic differentiation [21, 50]. This accumulative attacking strategy gives rise to a new threat for real-time systems, since the destruction happens only after a single update step before a monitor can perceive and intervene. Intuitively, the mechanism of the accumulative phase seems to be analogous to backdoor attacks [38, 54], while in Sec. 3.4 we discuss the critical differences between them.

Empirically, we conduct experiments on MNIST and CIFAR-10 by simulating different training processes encountered in two typical real-time streaming settings, involving online learning [8] and federated learning [30]. We demonstrate the effectiveness of accumulative poisoning attacks, and provide extensive ablation studies on different implementation details and tricks. We show that accumulative poisoning attacks can more easily bypass defenses like anomaly detection and gradient clipping than vanilla poisoning attacks. While previous efforts primarily focus on protecting the privacy of personal/client data [42, 44, 55], much less attention is paid to defend the integrity of the shared online or federated models. Our results advocate the necessity of embedding more robust defense mechanisms against poisoning attacks when learning from real-time data streaming.

## 2 Backgrounds

In this section, we will introduce three attacking strategies, and two typical paradigms of learning from real-time data streaming. For a classifier $f(x; \theta)$ with model parameters $\theta$, the training objective is denoted as $\mathcal{L}(x, y; \theta)$, where $(x, y)$ is the input-label pair. For notation compactness, we denote the empirical training objective on a data set or batch $S = \{x_i, y_i\}_{i=1}^N$ as $\mathcal{L}(S; \theta)$.

### 2.1 Attacking strategies

Below we briefly introduce the concepts of poisoning attacks [5], backdoor attacks [9], and adversarial attacks [19]. Although they may have different attacking goals, the applied techniques are similar, e.g., solving certain optimization problems by gradient-based methods.

**Poisoning attacks.** There is extensive prior work on poisoning attacks, especially in the offline settings against SVM [5], logistic regression [45], collaborative filtering [36], feature selection [65], and neural networks [13, 28, 29, 46, 56, 58]. In the threat model of poisoning attacks, the attacking goal is to degrade the model performance (e.g., test accuracy), while adversaries only have access to training data. Let $S_{\text{train}}$ be the clean training set and $S_{\text{val}}$ be a separate validation set, a poisoner will modify $S_{\text{train}}$ into a poisoned $\mathcal{P}(S_{\text{train}})$, and the malicious objective is formulated as

$$\max_{\mathcal{P}} \mathcal{L}\left(S_{\text{val}}; \theta^*\right), \text{ and } \theta^* \in \arg\min_{\theta} \mathcal{L}\left(\mathcal{P}(S_{\text{train}}); \theta\right), \tag{1}$$

where $S_{\text{train}}$ and $S_{\text{val}}$ are sampled from the same underlying distribution. The minimization problem is usually solved by stochastic gradient descent (SGD), where feeding data is randomized.

**Backdoor attacks.** As a variant of poisoning attacks, a backdoor attack aims to mislead the model on some specific target inputs [1, 16, 25, 54, 69], or inject trigger patterns [9, 22, 38, 52, 59, 61], without affecting model performance on clean test data. Backdoor attacks have a similar formulation as poisoning attacks, except that $S_{\text{val}}$ in Eq. (1) is sampled from a target distribution or with trigger patterns. Compared to the threat model of poisoning attacks, backdoor attacks assume additional accessibility in inference, where the test inputs could be specified or embedded with trigger patches.

**Adversarial attacks.** In recent years, adversarial vulnerability has been widely studied [12, 14, 19, 41, 47, 48, 60], where human imperceptible perturbations can be crafted to mislead image classifiers. Adversarial attacks usually only assume accessibility to test data. Under $\ell_p$-norm threat model, adversarial examples are crafted as

$$x^* \in \arg\max_{x'} \mathcal{L}(x', y; \theta), \text{ such that } \|x' - x\|_p \leq \epsilon, \tag{2}$$

where $\epsilon$ is the allowed perturbation size. In the adversarial literature, the constrained optimization problem in Eq. (2) is usually solved by projected gradient descent (PGD) [41].

## 2.2 Learning on real-time data streaming

This paper considers two typical real-time learning paradigms, i.e., online learning and federated learning, as briefly described below.

**Online learning.** Many practical services like recommendation systems rely on online learning [3, 8, 24] to update or refine their models, by exploiting the collected feedback from users or data in the wild. After capturing a new training batch $S_t$ at round $t$, the model parameters are updated as

$$\theta_{t+1} = \theta_t - \beta \nabla_\theta \mathcal{L}(S_t; \theta_t), \tag{3}$$

where $\beta$ is the learning rate of gradient descent. The optimizer can be more advanced (e.g., with momentum), while we use the basic form of gradient descent in our formulas to keep compactness.

**Federated learning.** Recently, federated learning [30, 43, 55] become a popular research area, where distributed devices (clients) collaboratively learn a shared prediction model, and the training data is kept locally on each client for privacy. At round $t$, the server distributes the current model $\theta_t$ to a subset $I_t$ of the total $N$ clients, and obtain the model update as

$$\theta_{t+1} = \theta_t - \beta \sum_{n \in I_t} G_t^n, \tag{4}$$

where $\beta$ is the learning rate and $G_t^1, \cdots, G_t^N$ are the updates potentially returned by the $N$ clients. Compared to online learning that captures semantic data (e.g., images), the model updates received in federated learning are non-semantic for a human monitor, and harder to execute anomaly detection.

## 3 Accumulative poisoning attacks

Conceptually, a vanilla online poisoning attack [63] greedily feeds the model with poisoned data, and a monitor could stop the training process after observing a gradual decline of model accuracy. In contrast, we propose accumulative poisoning attacks, where the model states are secretly (i.e., keeping accuracy in a reasonable range) activated towards a trigger batch by the accumulative phase, and the model is suddenly broken down by feeding in the trigger batch, before the monitor gets

conscious of the threat. During the accumulative phase, we need to calculate higher-order derivatives, while by using the reverse-mode automatic differentiation in modern libraries like PyTorch [50], the extra computational burden is usually constrained up to $2 \sim 5$ times more compared to the forward propagation [20, 21], which is still efficient. This section formally demonstrates the connections between a vanilla online poisoner and the accumulative phase and provides empirical algorithms for evading the models trained by online learning and federated learning, respectively.

## 3.1 Poisoning attacks in real-time data streaming

Recent work takes up research on poisoning attacks in real-time data streaming against online SVM [7], autoregressive models [2, 10], bandit algorithms [27, 37, 40], and classification [35, 63, 66]. In this paper, we focus on the classification tasks under real-time data streaming, e.g., online learning and federated learning, where the minimization process in Eq. (1) is usually substituted by a series of model update steps [11]. Assuming that the poisoned data/gradients are fed into the model at round $T$, then according to Eq. (3) and Eq. (4), the real-time poisoning problems can be formulated as

$$\max_{\mathcal{P}} \mathcal{L}\left(S_{\text{val}}; \theta_{T+1}\right), \text{ where } \theta_{T+1} = \begin{cases} \theta_T - \beta \nabla_\theta \mathcal{L}\left(\mathcal{P}(S_T); \theta_T\right), & \text{online learning;} \\ \theta_T - \beta \sum_{n \in I_T} \mathcal{P}(G_T^n), & \text{federated learning,} \end{cases} \quad (5)$$

where $I_T$ is the subset of clients selected at round $T$. The poisoning operation $\mathcal{P}$ acts on the data points in online learning, while acting on the gradients in federated learning. To quantify the ability of poisoning attackers, we define the poisoning ratio as $\mathcal{R}(\mathcal{P}, S) = \frac{|\mathcal{P}(S) \backslash S|}{|S|}$, where $S$ could be a set of data or model updates, and $|S|$ represents the number of elements in $S$.

Note that in Eq. (5), the poisoning operation $\mathcal{P}$ optimizes a shortsighted goal, i.e., greedily decreasing model accuracy at each update step, which limits the destructive effect induced by every poisoned batch, and only causes a gradual descent of accuracy. Moreover, this kind of poisoning behavior is relatively easy to perceive by a heuristic monitor [51], and set aside time for intervention before the model being irreparably broken. In contrast, we propose an accumulative poisoning strategy, which can be regarded as indirectly optimizing $\theta_T$ in Eq. (5) via a succession of updates, as detailed below.

## 3.2 Accumulative poisoning attacks in online learning

By expanding the online learning objective of Eq. (5) in first-order terms (up to an $\mathcal{O}(\beta^2)$ error), we can rewrite the maximization problem as

$$\max_{\mathcal{P}} \mathcal{L}\left(S_{\text{val}}; \theta_T\right) - \beta \nabla_\theta \mathcal{L}\left(S_{\text{val}}; \theta_T\right)^\top \nabla_\theta \mathcal{L}\left(\mathcal{P}(S_T); \theta_T\right)$$
$$\Rightarrow \min_{\mathcal{P}} \nabla_\theta \mathcal{L}\left(S_{\text{val}}; \theta_T\right)^\top \nabla_\theta \mathcal{L}\left(\mathcal{P}(S_T); \theta_T\right). \quad (6)$$

Notice that the vanilla poisoning attack only maliciously modifies $S_T$, while keeping the pretrained parameters $\theta_T$ uncontrolled. Motivated by this observation, a natural way to amplify the destructive effects (i.e., obtain a lower value for the minimization problem in Eq. (6)) is to jointly poison $\theta_T$ and $S_T$. Although we cannot directly manipulate $\theta_T$, we exploit the fact that the data points are captured sequentially. We inject an accumulative phase $\mathcal{A}$ to make $\mathcal{A}(\theta_T)$[1] be more sensitive to the clean batch $S_T$ or poisoned batch $\mathcal{P}(S_T)$, where we call $S_T$ or $\mathcal{P}(S_T)$ as the **trigger batch** in our methods. Based on Eq. (6), accumulative poisoning attacks can be formulated as

$$\min_{\mathcal{P}, \mathcal{A}} \nabla_\theta \mathcal{L}\left(S_{\text{val}}; \mathcal{A}(\theta_T)\right)^\top \nabla_\theta \mathcal{L}\left(\mathcal{P}(S_T); \mathcal{A}(\theta_T)\right), \quad (7)$$

where $\mathcal{L}\left(S_{\text{val}}; \mathcal{A}(\theta_T)\right) \leq \mathcal{L}\left(S_{\text{val}}; \theta_T\right) + \gamma$, and $\gamma$ is a hyperparameter controlling the tolerance on performance degradation in the accumulative phase, in order to bypass monitoring on model accuracy.

**Implementation of $\mathcal{A}$.** Now, we describe how to implement the accumulative phase $\mathcal{A}$. Assuming that the online process begins from a burn-in phase, resulting in $\theta_0$, and let $S_0, \cdots, S_{T-1}$ be the clean online data batches at rounds $0, \cdots, T-1$ after the burn-in phase. The accumulative phase iteratively trains the model on the perturbed data batch $\mathcal{A}_t(S_t)$, update the parameters as

$$\theta_{t+1} = \theta_t - \beta \nabla_\theta \mathcal{L}(\mathcal{A}_t(S_t); \theta_t). \quad (8)$$

---

[1]The notation $\mathcal{A}(\theta_T)$ refers to the model parameters at round $T$ obtained after the accumulative phase.

---

**Algorithm 1** Accumulative poisoning attacks in online learning

---

**Input:** Burn-in parameters $\theta_0$; training batches $S_t = \{x_i^t, y_i^t\}_{i=1}^N$, $t \in [0, T]$; validation batch $S_{\mathrm{val}}$.

Initialize $\mathcal{P}(S_T) = S_T$;

**for** $t = 0$ **to** $T-1$ **do**

    Initialize $\mathcal{A}_t(S_t) = S_t$;

    Bootstrap $S_{\mathrm{val}}$, and/or normalize $\nabla_\theta \mathcal{L}(S_t; \theta_t)$, $\nabla_\theta \mathcal{L}(S_T; \theta_t)$, $\nabla_\theta \mathcal{L}(S_{\mathrm{val}}, \theta_t)$;        *# optional*

    **for** $c = 1$ **to** $C$ **do**

        Compute $G_t = \nabla_\theta \left( \nabla_\theta \mathcal{L}(S_{\mathrm{val}}, \theta_t)^\top \nabla_\theta \mathcal{L}(S_T; \theta_t) \right)$;

        Compute $H_t = \nabla_\theta \mathcal{L}(S_t; \theta_t)^\top \left[ \nabla_\theta \mathcal{L}(S_t^\dagger; \theta_t) + \lambda \cdot G_t \right]$, where $\dagger$ stops gradients;

        Update $\mathcal{A}_t(x_i^t) = \mathrm{proj}_\epsilon \left( \mathcal{A}_t(x_i^t) + \alpha \cdot \mathrm{sign}(\nabla_{x_i^t} H_t) \right)$ for $i \in [1, N]$;        *# update $\mathcal{A}_t(S_t)$*

        Update $\mathcal{P}(x_i^T) = \mathrm{proj}_\epsilon \left( \mathcal{P}(x_i^T) + \alpha \cdot \mathrm{sign}(\nabla_{x_i^T} H_t) \right)$ for $i \in [1, N]$;        *# update $\mathcal{P}(S_T)$*

    **end for**

    Update $\theta_{t+1} = \theta_t - \beta \nabla_\theta \mathcal{L}(\mathcal{A}_t(S_t); \theta_t)$;        *# feed in $\mathcal{A}_t(S_t)$*

**end for**

Update $\theta_{T+1} = \theta_T - \beta \nabla_\theta \mathcal{L}(\mathcal{P}(S_T); \theta_T)$;        *# feed in $\mathcal{P}(S_T)$*

**Return:** The poisoned parameters $\theta_{T+1}$.

---

According to the malicious objective in Eq. (7) and the updating rule in Eq. (8), we can craft the perturbed data batch $\mathcal{A}(S_t)$ at round $t$ by solving (under first-order expansion)

$$\max_{\mathcal{P}, \mathcal{A}_t} \nabla_\theta \mathcal{L}(\mathcal{A}_t(S_t); \theta_t)^\top \left[ \nabla_\theta \mathcal{L}(S_t; \theta_t) + \lambda \cdot \nabla_\theta \left( \nabla_\theta \mathcal{L}(S_{\mathrm{val}}, \mathcal{A}(\theta_T))^\top \nabla_\theta \mathcal{L}(\mathcal{P}(S_T); \mathcal{A}(\theta_T)) \right) \right]$$

$$\Rightarrow \max_{\mathcal{P}, \mathcal{A}_t} \nabla_\theta \mathcal{L}(\mathcal{A}_t(S_t); \theta_t)^\top \left[ \underbrace{\nabla_\theta \mathcal{L}(S_t; \theta_t)}_{\text{keeping accuracy}} + \lambda \cdot \underbrace{\nabla_\theta \left( \nabla_\theta \mathcal{L}(S_{\mathrm{val}}, \theta_t)^\top \nabla_\theta \mathcal{L}(\mathcal{P}(S_T); \theta_t) \right)}_{\text{accumulating poisoning effects for the trigger batch}} \right], \tag{9}$$

where $t \in [0, T-1]$ (we abuse the notation $[a, b]$ to denote the set of integers from $a$ to $b$). Specifically, in the first line of Eq. (9), $\nabla_\theta \mathcal{L}(S_t; \theta_t)$ is the gradient on the clean batch $S_t$, and $\nabla_\theta \left( \nabla_\theta \mathcal{L}(S_{\mathrm{val}}, \mathcal{A}(\theta_T))^\top \nabla_\theta \mathcal{L}(\mathcal{P}(S_T); \mathcal{A}(\theta_T)) \right)$ is the gradient of the minimization problem in Eq. (7). Solving the maximization problem in Eq. (9) is to make the accumulative gradient $\nabla_\theta \mathcal{L}(\mathcal{A}_t(S_t); \theta_t)$ to align with $\nabla_\theta \mathcal{L}(S_t; \theta_t)$ and $\nabla_\theta \left( \nabla_\theta \mathcal{L}(S_{\mathrm{val}}, \mathcal{A}(\theta_T))^\top \nabla_\theta \mathcal{L}(\mathcal{P}(S_T); \mathcal{A}(\theta_T)) \right)$ simultaneously, with a trade-off hyperparameter $\lambda$. In the second line, since we cannot calculate $\mathcal{A}(\theta_T)$ in advance, we greedily approximate $\mathcal{A}(\theta_T)$ by $\theta_t$ in each accumulative step.

In Algorithm 1, we provide an instantiation of accumulative poisoning attacks in online learning. At the beginning of each round of accumulation, it is optional to bootstrap $S_{\mathrm{val}}$ to avoid overfitting, and normalize the gradients to concentrate on angular distances. When computing $H_t$, we apply stopping gradients (e.g., the detach operation in PyTorch [50]) to control the back-propagation flows.

**Capacity of poisoners.** To ensure that the perturbations are imperceptible for human observers, we follow the settings in the adversarial literature [19, 60], and constrain the malicious perturbations on data into $\ell_p$-norm bounds. The update rules of $\mathcal{P}$ and $\mathcal{A}_t$ are based on projected gradient descent (PGD) [41] under $\ell_\infty$-norm threat model, where iteration steps $C$ and step size $\alpha$, and maximal perturbation $\epsilon$ are hyperparameters. Other techniques like GANs [18] can also be applied to generate semantic perturbations, while we do not further explore them.

**Poisoning ratios.** The ratios of poisoned data have different meanings in online/real-time and offline settings. Namely, in real-time settings, we only poison data during the accumulative phase. If we ask *the ratio of poisoned data points that are fed into the model*, the formula should be

$$\frac{\text{Per-batch poisoning ratio} \times \text{Accumulative epochs}}{\text{Burin-in epochs} + \text{Accumulative epochs}}.$$

So for example in Fig. 1, even if we use $100\%$ per-batch poisoning ratio during the accumulative phase for 2 epochs, the ratio of poisoned data points fed into the model is only $100\% \times 2/(40+2) \approx 4.76\%$, where 40 is the number of burn-in epochs. In contrast, if we poison $10\%$ data in an offline dataset, then the expected ratio of poisoned data points fed into the model is also $10\%$.

---

**Algorithm 2** Accumulative poisoning attacks in federated learning

---

**Input:** Burn-in parameters $\theta_0$; training updates $\{G_t^n\}_{n=1}^N$, $t \in [0, T]$, where we assume that $G_T^n$ is computed by the local data batch $S_T^n$ as $G_T^n = \nabla_\theta \mathcal{L}(S_T^n; \theta_T)$; validation batch $S_{\text{val}}$.

**Input:** Sampled index sets $I_t$, where $t \in [0, T]$.                                    # *access to random seeds*

Initialize $\mathcal{P}(G_T^n) = \text{proj}_\eta(-\nabla_\theta \mathcal{L}(S_T^n; \theta_0))$ for $n \in I_T$;                       # *reverse trigger*

**for** $t = 0$ **to** $T - 1$ **do**

    Sample random vectors $\{M_t^n\}_{n=1}^N$ such that $\sum_{i=1}^N M_t^n = 0$;

    Initialize $\mathcal{A}_t(G_t^n) = M_t^n$ for $n \in I_t$;

    Bootstrap $S_{\text{val}}$, and/or normalize $\nabla_\theta \mathcal{L}(S_{\text{val}}, \theta_t)$;                        # *optional*

    Update $\mathcal{P}(G_T^n) = \text{proj}_\eta(\mathcal{P}(G_T^n) - \alpha \cdot \nabla_\theta \mathcal{L}(S_T^n; \theta_t))$ for $n \in I_T$;

    Compute $H_t = \sum_{n \in I_t} G_t^n + \lambda \cdot \nabla_\theta \left( \nabla_\theta \mathcal{L}(S_{\text{val}}; \theta_t)^\top \sum_{n \in I_t} \mathcal{P}(G_T^n) \right)$;

    Update $\mathcal{A}_t(G_t^n) = \text{proj}_\eta(\mathcal{A}_t(G_t^n) + H_t)$ for $n \in I_t$;

    Update $\theta_{t+1} = \theta_t - \beta \sum_{n \in I_t} \mathcal{A}_t(G_t^n)$;                       # *feed in* $\mathcal{A}_t(G_t^n)$, $n \in I_t$

**end for**

Update $\theta_{T+1} = \theta_T - \beta \sum_{n \in I_T} \mathcal{P}(G_T^n)$;                       # *feed in* $\mathcal{P}(G_T^n)$, $n \in I_T$

**Return:** The poisoned parameters $\theta_{T+1}$.

---

## 3.3 Accumulative poisoning attacks in federated learning

Similar as the derivations in Eq. (6) and Eq. (7), under first-order expansion, accumulative poisoning attacks in federated learning can be formulated by the minimization problem

$$\min_{\mathcal{P}, \mathcal{A}} \nabla_\theta \mathcal{L}(S_{\text{val}}; \mathcal{A}(\theta_T))^\top \sum_{n \in I_T} \mathcal{P}(G_T^n), \tag{10}$$

such that $\mathcal{L}(S_{\text{val}}; \mathcal{A}(\theta_T)) \leq \mathcal{L}(S_{\text{val}}; \theta_T) + \gamma$. Assuming that the federated learning process begins from a burn-in process, resulting in a shared model of parameters $\theta_0$, and then being induced into an accumulative phase from round 0 to $T - 1$. At round $t$, the accumulative phase updates the model as

$$\theta_{t+1} = \theta_t - \beta \sum_{n \in I_t} \mathcal{A}_t(G_t^n). \tag{11}$$

According to the formulations in Eq. (10) and Eq. (11), the perturbation $\mathcal{A}_t$ can be obtained by

$$\max_{\mathcal{P}, \mathcal{A}_t} \left( \sum_{n \in I_t} \mathcal{A}_t(G_t^n) \right)^\top \left[ \sum_{n \in I_t} G_t^n + \lambda \cdot \nabla_\theta \left( \nabla_\theta \mathcal{L}(S_{\text{val}}; \theta_t)^\top \sum_{n \in I_T} \mathcal{P}(G_T^n) \right) \right], \tag{12}$$

where $\lambda$ is a trade-off hyperparameter similar as in Eq. (9), and $\mathcal{A}(\theta_T)$ is substituted by $\theta_t$ at each round $t$. In Algorithm 2, we provide an instantiation of accumulative poisoning attacks in federated learning. The random vectors $M_t^n$ are used to avoid the model updates from different clients being the same (otherwise, a monitor will perceive the abnormal behaviors). We assume white-box access to random seeds, while black-box cases can also be handled [4]. Different from the case of online learning, we do not need to run $C$-steps PGD to update the poisoned trigger or accumulative batches.

**Capacity of poisoners.** Unlike online learning in which the captured data is usually semantic, the model updates received during federated learning are just numerical matrices or tensors, thus a human observer cannot easily perceive the malicious behaviors. If we allow $\mathcal{P}$ and $\mathcal{A}_t$ to be arbitrarily powerful, then it is trivial to break down the trained models. To make the settings more practical, we clip the poisoned gradients under $\ell_p$-norm, namely, we constrain that $\forall n \in I_T$ and $t \in [0, T-1]$, there are $\|\mathcal{P}(G_T^n)\|_p \leq \eta$ and $\|\mathcal{A}_t(G_t^n)\|_p \leq \eta$, where $\eta$ has a similar role as the perturbation size $\epsilon$ in adversarial attacks. Empirical results under gradient clipping can be found in Sec. 4.2.

**Reverse trigger.** When initializing the poisoned trigger, we apply a simple trick of reversing the clean trigger batch. Specifically, the gradient computed on a clean trigger batch $S_T$ at round $t$ is $\nabla_\theta \mathcal{L}(S_T; \theta_t)$. A simple way to increase the validate loss value is to reverse the model update to be $-\nabla_\theta \mathcal{L}(S_T; \theta_t)$, which convert the accumulative objective in Eq. (10) as

$$\min_{\mathcal{A}_t} \nabla_\theta \mathcal{L}(S_T; \mathcal{A}_t(\theta_t))^\top \nabla_\theta \mathcal{L}(S_{\text{val}}; \mathcal{A}_t(\theta_t)) \implies \max_{\mathcal{A}_t} \nabla_\theta \mathcal{L}(S_T; \mathcal{A}_t(\theta_t))^\top \nabla_\theta \mathcal{L}(S_{\text{val}}; \mathcal{A}_t(\theta_t)), \tag{13}$$

Table 1: Classification accuracy (%) of the simulated online learning models on CIFAR-10. The default settings: ratio $\mathcal{R} = 100\%$, and the poisoned trigger $\mathcal{P}$ is fixed during the process of accumulative phase. We perform ablation studies on different tricks used in the accumulative phase.

| Method | Acc. before trigger | Acc. after trigger | $\Delta$ |
|---|---|---|---|
| Clean trigger | 83.38 | 84.07 | $+0.69$ |
| **+ accumulative phase** | $80.90\pm0.50$ | $76.94\pm0.89$ | $-3.95\pm0.61$ |
| + re-sampling $S_{\mathrm{val}}$ | $80.69\pm0.34$ | $76.65\pm0.93$ | $-4.03\pm0.66$ |
| + weight momentum | $78.39\pm0.94$ | $70.17\pm1.50$ | $\mathbf{-8.23\pm0.88}$ |
| Poisoned trigger | | | |
| + $\epsilon = 8/255$ | 83.38 | 82.11 | $-1.27$ |
| **+ accumulative phase** | $81.37\pm0.12$ | $78.06\pm0.68$ | $-3.31\pm0.57$ |
| + re-sampling $S_{\mathrm{val}}$ | $80.45\pm0.25$ | $78.18\pm0.84$ | $-3.27\pm0.62$ |
| + weight momentum | $81.47\pm0.50$ | $77.11\pm0.38$ | $-4.36\pm0.44$ |
| + optimizing $\mathcal{P}$ | $81.31\pm0.33$ | $76.05\pm0.40$ | $-5.26\pm0.33$ |
| + weight momentum | $80.77\pm1.00$ | $74.05\pm1.20$ | $\mathbf{-6.72\pm0.70}$ |
| + $\epsilon = 16/255$ | 83.38 | 80.85 | $-2.53$ |
| **+ accumulative phase** | $81.43\pm0.17$ | $77.89\pm0.82$ | $-3.54\pm0.96$ |
| + re-sampling $S_{\mathrm{val}}$ | $81.61\pm0.11$ | $77.87\pm0.79$ | $-3.74\pm0.69$ |
| + weight momentum | $80.57\pm0.12$ | $74.82\pm1.00$ | $-5.75\pm1.08$ |
| + optimizing $\mathcal{P}$ | $80.02\pm0.92$ | $71.10\pm1.68$ | $-8.92\pm0.77$ |
| + weight momentum | $80.17\pm1.24$ | $69.08\pm1.72$ | $\mathbf{-11.09\pm0.57}$ |
| + $\epsilon = 0.1$ | 83.38 | 80.52 | $-2.86$ |
| **+ accumulative phase** | $81.20\pm0.14$ | $74.29\pm0.21$ | $-6.91\pm0.17$ |
| + re-sampling $S_{\mathrm{val}}$ | $81.43\pm0.41$ | $74.73\pm0.82$ | $-6.70\pm0.98$ |
| + weight momentum | $79.46\pm0.56$ | $69.90\pm1.01$ | $-9.56\pm0.77$ |
| + optimizing $\mathcal{P}$ | $81.16\pm0.57$ | $70.13\pm0.88$ | $-11.04\pm0.56$ |
| + weight momentum | $81.34\pm0.15$ | $69.35\pm1.42$ | $\mathbf{-11.99\pm1.27}$ |

Figure 2: Metric values w.r.t. perturbation sizes under four anomaly detection methods, where lower metric values indicate outliers. The simulated online learning trains the model on CIFAR-10.

where the later objective is easier to optimize since for a burn-in model, the directions of $\nabla_\theta \mathcal{L}(S_T; \theta_t)$ and $\nabla_\theta \mathcal{L}(S_{\mathrm{val}}; \theta_t)$ are approximately aligned, due to the generalization guarantee. This trick can maintain the gradient norm unchanged, and does not exploit the validation batch for training.

**Recovered offset.** Let $G_t = \{G_t^n\}_{n \in I_t}$ be the gradient set received at round $t$, then if we can only modify a part of these gradients, i.e., $\mathcal{R}(\mathcal{A}_t, G_t) < 1$, we can apply a simple trick to recover the original optimal solution. Technically, assuming that we can only modify the clients in $I_t' \subset I_t$, and the optimal solution of $\mathcal{A}_t$ in Eq. (12) is $\mathcal{A}_t^*$, then we can modify the clients according to

$$\sum_{n \in I_t'} \mathcal{A}_t(G_t^n) = \sum_{n \in I_t} \mathcal{A}_t^*(G_t^n) - \sum_{n \in I_t \setminus I_t'} G_t^n, \text{ where } \mathcal{R}(\mathcal{A}_t, G_t) = \frac{|I_t'|}{|I_t|}. \tag{14}$$

The trick shown in Eq. (14) can help us to eliminate the influence of unchanged model updates, and stabilize the update directions to follow the malicious objective during the accumulative phase.

## 3.4 Differences between the accumulative phase and backdoor attacks

Although both the accumulative phase and backdoor attacks [22, 38] can be viewed as making the model be sensitive to certain trigger batches, there are critical differences between them:

Table 2: Classification accuracy (%) on CIFAR-10 by setting different data poisoning ratios in online learning. The results are of fixing the poisoned trigger and optimizing it in the accumulative phase.

| Method | | | Ratio (%) | | | | | | | | |
|---|---|---|---|---|---|---|---|---|---|---|---|
| | | 100 | 90 | 80 | 70 | 60 | 50 | 40 | 30 | 20 | 10 |
| **Accumulative phase** | Before | 81.64 | 81.49 | 80.03 | 81.02 | 81.06 | 81.57 | 81.60 | 81.90 | 81.35 | 81.43 |
| + Poisoned trigger $\mathcal{P}$ | After | 74.94 | 74.11 | 74.66 | 76.10 | 77.04 | 78.46 | 78.65 | 79.79 | 79.46 | 79.28 |
| | $\Delta$ | 6.67 | 7.38 | 5.37 | 4.92 | 4.02 | 3.11 | 2.95 | 2.11 | 1.89 | 2.15 |
| **Accumulative phase** | Before | 77.98 | 79.34 | 80.30 | 81.82 | 78.54 | 79.39 | 81.31 | 79.73 | 81.90 | 81.37 |
| + Optimizing $\mathcal{P}$ | After | 65.95 | 67.64 | 68.21 | 71.83 | 66.14 | 71.14 | 73.86 | 73.25 | 76.41 | 75.14 |
| | $\Delta$ | 12.03 | 11.70 | 12.09 | 9.99 | 12.40 | 8.25 | 7.45 | 6.48 | 5.49 | 6.23 |

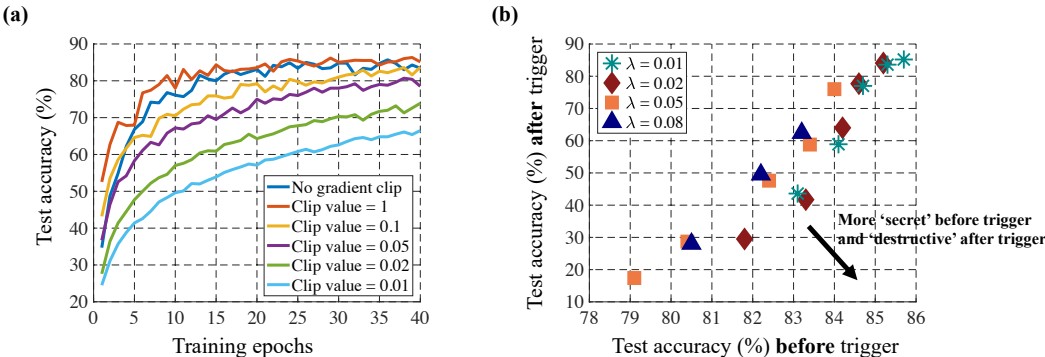

Figure 3: **(a)** The negative effect of lowing down the convergence rate of model training, when we apply gradient clipping to defend poisoning attacks. **(b)** Ablation studies on the value of $\lambda$ in Eq. (12).

**(i) Data accessibility and trigger occasions.** Backdoor attacks require accessibility on both training and test data, while our methods only need to manipulate training data. Besides, backdoor attacks trigger the malicious behaviors (e.g., fooling the model on specific inputs) during inference, while in our methods the malicious behaviors (e.g., breaking down the model accuracy) are triggered during training. **(ii) Malicious objectives.** We generally denote a trigger batch as $S_{\text{tri}}$. For backdoor attacks, the malicious objective can be formulated as $\max_{\mathcal{B}} \mathcal{L}(S_{\text{tri}}; \mathcal{B}(\theta))$, where $\mathcal{B}$ is the backdoor operations. In contrast, our accumulative phase optimizes $\min_{\mathcal{A}} \nabla_{\theta} \mathcal{L}(S_{\text{val}}; \mathcal{A}(\theta))^{\top} \nabla_{\theta} \mathcal{L}(S_{\text{tri}}; \mathcal{A}(\theta))$.

## 4 Experiments

We mimic the real-time data training using the MNIST and CIFAR-10 datasets [31, 33]. The learning processes are similar to regular offline pipelines, while the main difference is that the poisoning attackers are allowed to intervene during training and have access to the model states to tune their attacking strategies dynamically. Following [49], we apply ResNet18 [23] as the model architecture, and employ the SGD optimizer with momentum of 0.9 and weight decay of $1 \times 10^{-4}$. The initial learning rate is 0.1, and the mini-batch size is 100. The pixel values of images are scaled to be in the interval of 0 to 1.[2]

**Burn-in phase.** For all the experiments in online learning and federated learning, we pre-train the model for 10 epochs on the clean training data of MNIST, and 40 epochs on the clean training data of CIFAR-10. The learning rate is kept as 0.1, and the batch normalization (BN) layers [26] are in train mode to record feature statistics.

**Poisoning target.** Usually, when training on real-time data streaming, there will be a background process to monitor the model accuracy or recall, such that the training progress would be stopped if the monitoring metrics have significant retracement. Thus, we exploit single-step drop as a threatening poisoning target, namely, the accuracy drop after the model is trained by a single step with poisoned behaviors (e.g., trained on a batch of poisoned data). Single-step drop can measure the destructive effect caused by poisoning strategies, before monitors can react to the observed retracement.

---

[2]Code is available at `https://github.com/ShawnXYang/AccumulativeAttack`.

Table 3: Classification accuracy (%) on CIFAR-10 after the model updating by the trigger batch. The accumulative phase runs for 1,000 steps. Our methods better bypass the gradient clipping operations.

| Method | Loss scaling | No clip | $\ell_2$-norm clip bound | | | $\ell_\infty$-norm clip bound | | |
|---|---|---|---|---|---|---|---|---|
| | | | 10 | 1 | 0.1 | 10 | 1 | 0.1 |
| Poisoned trigger | 1 | 83.32 | 83.32 | 83.39 | 83.68 | 82.96 | 83.32 | 83.32 |
| | 10 | 65.28 | 70.16 | 83.14 | 83.66 | 65.28 | 68.04 | 82.07 |
| | 20 | 41.12 | 72.07 | 83.39 | 83.68 | 37.10 | 48.26 | 82.95 |
| | 50 | **10.18** | 72.07 | 83.14 | 83.66 | **10.18** | 42.49 | 82.95 |
| **Accumulative phase** + Clean trigger | 0.01 | 33.84 | 33.84 | 74.00 | 82.72 | 33.84 | 43.62 | 75.12 |
| | 0.02 | 21.73 | 27.66 | 69.54 | 80.98 | 21.73 | 38.37 | 74.78 |
| | 0.05 | 12.64 | 25.42 | 63.47 | 78.98 | 12.64 | 35.02 | 70.57 |
| | 0.08 | 11.17 | **21.17** | **61.87** | **76.55** | 11.17 | **21.17** | **64.31** |

Table 4: Results when using longer burn-in phase on CIFAR-10 (i.e., running the burn-in phase for 100 epochs, compared to 40 epochs in Table 3).

| Method | Loss scaling | No clip | $\ell_\infty$-norm clip bound | | |
|---|---|---|---|---|---|
| | | | 10 | 1 | 0.1 |
| Poisoned trigger | 1 | 89.34 | 89.34 | 89.91 | 89.99 |
| | 10 | 45.29 | 84.45 | 89.91 | 89.99 |
| | 20 | 16.62 | 84.45 | 89.91 | 89.99 |
| | 50 | 10.24 | 84.45 | 89.91 | 89.99 |
| **Accu. phase** | 0.01 | 80.35 | 80.35 | 80.35 | 83.32 |
| | 0.02 | 25.45 | 25.45 | 25.45 | 76.06 |
| | 0.05 | 12.03 | 12.03 | 15.53 | 70.00 |
| | 0.08 | 11.07 | 11.07 | 14.23 | 64.74 |

Table 5: Classification accuracy (%) on MNIST after the model updating by the trigger batch. The accumulative phase runs for 200 steps ($\frac{1}{3}$ epochs), with perturbation constraint $\epsilon = 16/255$ and step size $\alpha = 2/255$.

| Method | Loss scaling | No clip | $\ell_\infty$-norm clip bound | | |
|---|---|---|---|---|---|
| | | | 10 | 1 | 0.1 |
| Poisoned trigger | 1 | 98.27 | 98.27 | 98.27 | 98.28 |
| | 10 | 95.49 | 95.49 | 98.12 | 98.28 |
| | 20 | 84.09 | 89.24 | 98.12 | 98.28 |
| | 50 | 31.93 | 89.24 | 98.12 | 98.28 |
| **Accu. phase** | 0.08 | 22.49 | 22.49 | 32.87 | 51.28 |

## 4.1 Performance in the simulated experiments of online learning

At each step of the accumulative phase in online learning, we obtain a training batch from the ordered data streaming, and craft accumulative poisoning examples. The perturbations are generated by PGD [41], and restricted to some feasible sets under $\ell_\infty$-norm. To keep the accumulative phase being 'secret', we will early-stop the accumulative procedure if the model accuracy becomes lower than a specified range (e.g., 80% for CIFAR-10). We evaluate the effects of the accumulative phase in online learning using a clean trigger batch and poisoned trigger batches with different perturbation budgets $\epsilon$. We set the number of PGD iterations as $C = 100$, and the step size is $\alpha = 2 \cdot \epsilon / C$.

Table 1 reports the test accuracy before and after the model is updated on the trigger batch. The single-step drop caused by the vanilla poisoned trigger is not significant, while after combining with our accumulative phase, the poisoner can make much more destructive effects. As seen, we obtain prominent improvements by introducing two simple techniques for the accumulative phase, including *weight momentum* which lightly increases the momentum factor as 1.1 (from 0.9 by default) of the SGD optimizer in accumulative phase, and *optimizing $\mathcal{P}$* that constantly updates adversarial poisoned trigger in the accumulative phase. Besides, we re-sample different $S_{\text{val}}$ to demonstrate a consistent performance in the setting named *re-sampling $S_{val}$*. We also study the effectiveness of setting different data poisoning ratios, as summarized in Table 2. To mimic more practical settings, we also do simulated experiments on the models using group normalization (GN) [64], as detailed in Appendix A.1. In Fig. 4 we visualize the training data points used during the burn-in phase (clean) and used during the accumulative phase (perturbed under $\epsilon = 16/255$ constraint). As observed, the perturbations are hardly perceptible, while we provide more instances in Appendix A.2.

**Anomaly detection.** We evaluate the performance of our methods under anomaly detection. Kernel density (KD) [15] applies a Gaussian kernel $K(z_1, z_2) = \exp(-\|z_1 - z_2\|_2^2/\sigma^2)$ to compute the similarity between two features $z_1$ and $z_2$. For KD, we restore $1,000$ correctly classified training features in each class and use $\sigma = 10^{-2}$. Local intrinsic dimensionality (LID) [39] applies $K$ nearest neighbors to approximate the dimension of local data distribution. We restore a total of $10,000$ correctly classified training data points, and set $K = 600$. We also consider two model-based

Table 6: The effects of clean and poisoned trigger batch used with **accumulative phase** on CIFAR-10. For example, using a poisoned trigger and accumulating 200 steps leads to a more destructive effect than using a clean trigger for 500 steps.

| Loss scaling | Trigger batch | Accumulative steps $T$ | | | |
|---|---|---|---|---|---|
| | | 50 | 100 | 200 | 500 |
| 0.01 | Clean | 85.17 | 83.52 | 76.96 | 58.83 |
| | Poisoned | **78.86** | **67.00** | **58.12** | **26.40** |
| 0.02 | Clean | 84.12 | 77.69 | 63.02 | 41.71 |
| | Poisoned | **68.68** | **61.92** | **34.36** | **15.59** |

Figure 4: Visualization of the training data points used in the simulated process of online leaning.

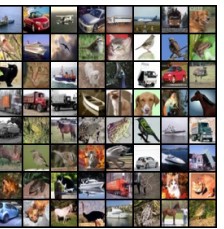
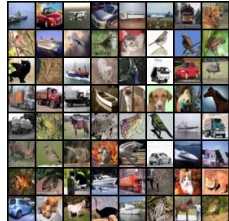

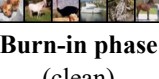

**Burn-in phase**      **Accumulative phase**
(clean)            ($\varepsilon = 16/255$)

detection methods, involving Gaussian mixture model (GMM) [67] and Gaussian discriminative analysis (GDA) [34]. In Fig. 2, the metric value for KD is the kernel density, for LID is the negative local dimension, for GDA and GMM is the likelihood. As observed, the accumulative poisoning samples can better bypass the anomaly detection methods, compared to the samples crafted by the vanilla poisoning attacks.

## 4.2 Performance in the simulated experiments of federated learning

In the experiments of federated learning, we simulate the model updates received from clients by computing the gradients on different mini-batches, following the setups in Konečný et al. [30], and we synchronize the statistics recorded by local BN layers. We first evaluate the effect of the accumulative phase along, by using a clean trigger batch. We apply the recovered offset trick (described in Eq. (14)) to eliminate potential influences induced by limited poisoning ratio, and perform ablation studies on the values of $\lambda$ in Eq. (12). As seen in Fig. 3 (b), we run the accumulative phase for different numbers of steps under each value of $\lambda$, and report the test accuracy before and after the model is updated on the trigger batch. As intuitively indicated, a point towards the bottom-right corner implies more secret before trigger batch (i.e., less accuracy drop and not easy to be perceived by monitor), while more destructive after the trigger batch. We can find that a modest value of $\lambda = 0.02$ performs well.

In Table 3, Table 4, and Table 5, we show that the accumulative phase can mislead the model with smaller norms of gradients, which can better bypass the clipping operations under both $\ell_2$ and $\ell_\infty$ norm cases. In contrast, previous strategies of directly poisoning the data batch to degrade model accuracy would require a large magnitude of gradient norm, and thus is easy to be defended by gradient clipping. On the other hand, executing gradient clipping is not for free, since it will lower down the convergence of model training, as shown in Fig. 3 (a). Finally, in Table 6, we show that exploiting poisoned trigger batch can further improve the computational efficiency of the accumulative phase, namely, using fewer accumulative steps to achieve a similar accuracy drop.

## 5 Conclusion

This paper proposes a new poisoning strategy against real-time data streaming by exploiting an extra accumulative phase. Technically, the accumulative phase secretly magnifies the model's sensitivity to a trigger batch by sequentially ordered accumulative steps. Our empirical results show that accumulative poisoning attacks can cause destructive effects by a single update step, before a monitor can perceive and intervene. We also consider potential defense mechanisms like different anomaly detection methods and gradient clipping, where our methods can better bypass these defenses and break down the model performance. These results can inspire more real-time poisoning strategies, while also appeal to strong and efficient defenses that can be deployed in practical online systems.

## Acknowledgements

This work was supported by the National Key Research and Development Program of China (Nos. 2020AAA0104304, 2017YFA0700904), NSFC Projects (Nos. 61620106010, 62061136001, 61621136008, 62076147, U19B2034, U19A2081, U1811461), Beijing Academy of Artificial Intelligence (BAAI), Tsinghua-Huawei Joint Research Program, a grant from Tsinghua Institute for Guo Qiang, Tsinghua University-China Mobile Communications Group Co.,Ltd. Joint Institute, Tiangong Institute for Intelligent Computing, and the NVIDIA NVAIL Program with GPU/DGX Acceleration.

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
