# A More experiments and technical details

We provide more empirical results and technical details to support our conclusions in the main text.

## A.1 Model architectures with group normalization (GN)

Table 1: Classification accuracy (%) of the simulated online learning models by using model architectures with group normalization (GN) on CIFAR-10 (we substitute BN layers in ResNet-18 with GN layers). We perform ablation studies on different tricks used in the accumulative phase.

| Method | Acc. before trigger | Acc. after trigger | $\Delta$ |
|---|---|---|---|
| Clean trigger | 82.99 | 83.58 | +0.59 |
| + **accumulative phase** | 80.84±1.08 | 76.16±0.76 | −4.68±0.78 |
| + weight momentum | 80.70±0.44 | 73.35±2.09 | **−7.36±1.70** |
| Poisoned trigger | | | |
| + $\epsilon = 8/255$ | 82.99 | 79.33 | −3.66 |
| + **accumulative phase** | 81.82±0.16 | 77.04±0.19 | −4.78±0.34 |
| + weight momentum | 81.35±0.81 | 75.40±0.75 | −5.95±0.60 |
| + optimizing $\mathcal{P}$ | 80.00±0.98 | 75.26±0.74 | −5.74±0.33 |
| + weight momentum | 80.51±1.48 | 73.50±1.70 | **−7.02±0.22** |
| + $\epsilon = 16/255$ | 82.99 | 78.46 | −4.53 |
| + **accumulative phase** | 81.49±0.39 | 76.19±0.94 | −5.30±0.56 |
| + weight momentum | 81.05±0.99 | 73.44±1.72 | −7.60±0.83 |
| + optimizing $\mathcal{P}$ | 80.75±0.84 | 72.75±1.31 | −8.00±1.24 |
| + weight momentum | 80.62±0.80 | 71.52±1.69 | **−9.10±1.34** |
| + $\epsilon = 0.1$ | 82.99 | 77.61 | −5.38 |
| + **accumulative phase** | 80.76±0.35 | 72.57±1.06 | −8.20±0.72 |
| + weight momentum | 80.40±0.75 | 70.64±1.64 | −9.76±0.90 |
| + optimizing $\mathcal{P}$ | 80.05±0.47 | 70.49±1.31 | −9.56±1.14 |
| + weight momentum | 80.05±0.47 | 68.88±1.42 | **−11.17±1.27** |

## A.2 Visualization of perturbed images

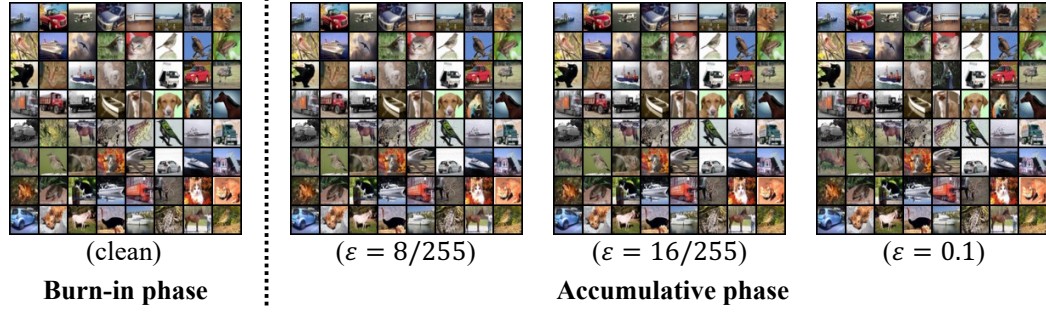

| (clean) | ($\varepsilon = 8/255$) | ($\varepsilon = 16/255$) | ($\varepsilon = 0.1$) |
|---|---|---|---|
| **Burn-in phase** | | **Accumulative phase** | |

Figure 1: We provide visualization on the perturbed accumulative poisoning samples under $\epsilon = 8/255$, $\epsilon = 16/255$, and $\epsilon = 0.1$, respectively. As seen, the crafted adversarial noises are hardly perceptible, especially in large-scale scenes that a human observer cannot easily distinguish the noise patterns.

## A.3 Technical details

Our methods are implemented by Pytorch [41], and run on GeForce RTX 2080 Ti GPU workers. The experiments of ResNet-18 are run by a single GPU. We assume that the poisoning adversaries have white-box accessibility to the model states, including the random seeds in the federated learning case. The CIFAR-10 dataset [23] consists of 60,000 32x32 colour images in 10 classes, with 6,000 images

per class. There are 50,000 training images and 10,000 test images. We perform RandomCrop with four paddings and RandomHorizontalFlip in training as the data augmentation.

**Computational complexity.** Empirically, we set the mini-batch as 100, and use 10-steps PGD attacks to execute poisoning. The running time for the vanilla poisoning attack is 2.33 seconds per batch, and for our accumulative poisoning attack is 2.47 seconds per batch.

# B  More backgrounds

This section introduces more backgrounds on poisoning attacks and backdoor attacks, and details on the adversarial attacks that we use to craft accumulative poisoning samples in our methods. Finally, we describe the commonly used anomaly detection methods against adversarially crafted samples, following previous settings [40].

## B.1  Poisoning attacks and backdoor attacks

There is extensive prior work on poisoning attacks, especially in the offline settings against SVM [3], logistic regression [36], collaborative filtering [27], feature selection [54], clustering [8], and neural networks [9, 21, 22, 38, 50]. Poisoning attacks in real-time data streaming are studied on online SVM [4], autoregressive models [1, 7], bandit algorithms [20, 31, 33], and classification [26, 52, 57].

Compared to poisoning attacks, backdoor attacks draw attention in more recent researches. These progresses involve backdoor attacks on self-supervised learning [42], point clouds [29, 51, 53], language models [28], graph neural networks [55], real physical world [56], brain computers [37], geenrative models [44], and image classification [14, 43, 45].

## B.2  Adversarial attacks

In online learning case, the poisoning capability is constrained under $\ell_p$-bounded threat models, where the perturbation $\delta$ is required to be bounded by a preset threshold $\epsilon$ under $\ell_p$-norm, i.e., $\|\delta\|_p \leq \epsilon$. Below we introduce the details of several adversarial attacks that can be used in our experiments.

**Fast gradient sign method (FGSM)** [16] generates an adversarial example under the $\ell_\infty$ norm as

$$\boldsymbol{x}^{adv} = \boldsymbol{x} + \epsilon \cdot \text{sign}(\nabla_{\boldsymbol{x}} \mathcal{L}_{\text{A}}(\boldsymbol{x}, y)), \tag{1}$$

where $\boldsymbol{x}$ is the original clean input, $y$ is the input label, $\boldsymbol{x}^{adv}$ is the crafted adversarial input, and $\mathcal{L}_{\text{A}}$ is the adversarial objective. The sign function $\text{sign}$ is used according to the first-order approximation under $\ell_\infty$-norm [39, 48].

**Projected gradient descent (PGD)** [34] extends FGSM by iteratively taking multiple small gradient updates as

$$\boldsymbol{x}_{t+1}^{adv} = \text{clip}_{\boldsymbol{x},\epsilon}\big(\boldsymbol{x}_t^{adv} + \eta \cdot \text{sign}(\nabla_{\boldsymbol{x}} \mathcal{L}_{\text{A}}(\boldsymbol{x}_t^{adv}, y))\big), \tag{2}$$

where $\text{clip}_{\boldsymbol{x},\epsilon}$ projects the adversarial example to satisfy the $\ell_\infty$ constraint and $\eta$ is the step size. Note that PGD involves a random initialization step as $\boldsymbol{x}_0^{adv} \sim \mathcal{U}(\boldsymbol{x} - \epsilon, \boldsymbol{x} + \epsilon)$.

**Momentum iterative method (MIM)** [10] integrates a momentum term into basic iterative method (BIM) [24] with the decay factor $\mu = 1.0$ as

$$\boldsymbol{g}_{t+1} = \mu \cdot \boldsymbol{g}_t + \frac{\nabla_{\boldsymbol{x}} \mathcal{L}_{\text{A}}(\boldsymbol{x}_t^{adv}, y)}{\|\nabla_{\boldsymbol{x}} \mathcal{L}_{\text{A}}(\boldsymbol{x}_t^{adv}, y)\|_1}, \tag{3}$$

where the adversarial examples are updated by

$$\boldsymbol{x}_{t+1}^{adv} = \text{clip}_{\boldsymbol{x},\epsilon}(\boldsymbol{x}_t^{adv} + \alpha \cdot \text{sign}(\boldsymbol{g}_{t+1})). \tag{4}$$

MIM has good performance as a transfer-based attack in the black-box setting.

## B.3  Anomaly detection in the adversarial setting

Recently, many defense methods are proposed against poisoning attacks [5, 6, 15, 35, 46, 47, 49] and against backdoor attacks [2, 11, 13, 17–19, 30]. Since we apply adversarial attacking methods

(i.e., PGD) to craft accumulative poisoning samples, we exploit related detection methods in the adversarial literature.

**Kernel density (KD).** In Feinman et al. [12], KD applies a Gaussian kernel $K(z_1, z_2) = \exp(-\|z_1 - z_2\|_2^2/\sigma^2)$ to compute the similarity between two features $z_1$ and $z_2$. There is a hyperparameter $\sigma$ controlling the bandwidth of the kernel, i.e., the smoothness of the density estimation. For KD, we restore $1,000$ correctly classified training features in each class and use $\sigma = 10^{-2}$.

**Local intrinsic dimensionality (LID).** In Ma et al. [32], LID applies $K$ nearest neighbors to approximate the dimension of local data distribution. The empirical estimation of LID is calculated as

$$\mathrm{LID}(x) = -\left( \frac{1}{K} \sum_{i=1}^{K} \log \frac{r_i}{r_K} \right)^{-1}, \tag{5}$$

where $r_i$ is the distance from $x$ to its $i$-th nearest neighbor. Note that we actually apply negative LID (i.e., $-\mathrm{LID}(x)$) to make sure that a lower metric value indicates outliers. Instead of computing LID in each mini-batch, we allow the detector to use a total of $10,000$ correctly classified training data points, and treat the number of $K$ as a hyperparameter. For LID, we restore a total of $10,000$ correctly classified training features and use $K = 600$.

**Gaussian-based detection.** Gaussian mixture model (GMM) [58] and Gaussian discriminative analysis (GDA) [25] are two commonly used Gaussian-based detection methods. Both of them fit a mixture of Gaussian model in the feature space of trained models. The main difference is that GDA uses all-classes data to fit a covariance matrix, and GMM fits conditional covariance matrices. We calculate the mean and covariance matrix on all correctly classified training samples.

## C    The capacity of the poisoning attackers

In the federated learning cases, we denote $G$ as the aggregated gradient from the clients, and $G_n$ as the gradient from the $n$-th client, where $n \in [1, N]$. We considered three threat models in our paper:

### C.1    No gradient clip: $G = \sum_n G_n$

In this case, we can only modify a *single* client, e.g., the $k$-th client, to achieve arbitrary poisoned aggregated gradient $G^{\mathrm{poi}}$. Specifically, by the simple trick of recovered offset, we poison $G_j$ to be $\mathcal{A}(G_j)$, where

$$\mathcal{A}(G_j) = G^{\mathrm{poi}} - \sum_{n \neq j} G_n,$$

such that $\mathcal{A}(G_j) + \sum_{n \neq j} G_n = G^{\mathrm{poi}}$. In our simulation experiments, we set batch size be 100, and treat each data point as a client. So we can only poison the gradient on a singe data point/client, i.e., the poisoning ratio is $1\%$. Ideally, in this case the poison ratio

$$\frac{1}{\text{number of clients}}$$

can be arbitrarily close to zero for large number of clients. The empirical results correspond to the "No clip" column.

### C.2    Gradient clip after aggregation: $G = \mathbf{Clip}_\eta(\sum_n G_n)$

Similar to the derivations above, in this case we can still poison a *single* client, e.g., the $j$-th client, such that

$$\mathbf{Clip}_\eta \left( \mathcal{A}(G_j) + \sum_{n \neq j} G_n \right) = \mathbf{Clip}_\eta \left( G^{\mathrm{poi}} \right),$$

where the poisoning ratio is still $1\%$ in our simulation experiments, and can be arbitrarily close to zero as discussed above. The empirical results correspond to the "$\ell_2$-norm clip bound" and "$\ell_\infty$-norm clip bound" columns.

## C.3 Gradient clip before aggregation: $G = \sum_n \textbf{Clip}_\eta(G_n)$

In this case, assuming that we can modify $M$ clients in the index set $S$, i.e., $|S| = M$ and the per-batch poison ratio is $\frac{M}{N}$. Specifically, for any $m \in S$, we poison $G_m$ to be $\mathcal{A}_m(G_m)$. Under the gradient clip constraint, we want to achieve $G^{\text{poi}}$ as well as we can, so we optimize the objective

$$\min_{\mathcal{A}_m, m \in S} \left\| G^{\text{poi}} - \left( \sum_{m \in S} \textbf{Clip}_\eta \left( \mathcal{A}_m(G_m) \right) + \sum_{n \notin S} \textbf{Clip}_\eta \left( G_n \right) \right) \right\|.$$

Under the mild condition that $\eta$ is small, the optimal solution of the above objective is $\forall m \in S$,

$$\mathcal{A}_m(G_m) = G^{\text{poi}} - \sum_{n \notin S} \textbf{Clip}_\eta \left( G_n \right).$$

and

$$\sum_{m \in S} \textbf{Clip}_\eta \left( \mathcal{A}_m(G_m) \right) = \sum_{m \in S} \textbf{Clip}_\eta \left( G^{\text{poi}} - \sum_{n \notin S} \textbf{Clip}_\eta \left( G_n \right) \right) = \textbf{Clip}_{M\eta} \left( G^{\text{poi}} - \sum_{n \notin S} \textbf{Clip}_\eta \left( G_n \right) \right).$$

As we can see, poisoning more clients (i.e., larger $M$) can be regarded as relaxing the gradient clip constraint (i.e., relax from $\eta$ to $M\eta$).