# OpenReview forum: "Accumulative Poisoning Attacks on Real-time Data"
_NeurIPS.cc/2021/Conference — NeurIPS 2021 Poster_

### Official Review · Reviewer_C5BF · 2021-07-16

**Rating:** 7
**Confidence:** 3

**Summary:**

This paper proposes a poisoning attack in settings where data is acquired iteratively. Two concrete settings are examined: online learning and federated learning. Under the assumed threat model, an attacker may observe the state of the learning algorithm and poison the data it consumes in each iteration. In order to evade detection, a novel attack strategy is proposed where the learned model is primed over several iterations in a so-called _accumulative phase_ without causing harm. After the accumulative phase, the attack can be activated by a special trigger data batch to cause a sudden drop in model accuracy. The attack is formulated as an optimization problem, which is ultimately relaxed and solved using projected gradient descent. Experiments are performed using CIFAR-10 to understand the power of the attack for different parameter settings and variations of the accumulative phase. The results show that the attack is effective and can more easily bypass anomaly detection and gradient clipping defenses.

**Limitations And Societal Impact:**

Societal impact is not discussed at length, but new types of attacks are usually disclosed so that defenses can be developed.

**Main Review:**

This paper makes a valuable contribution to the literature on data poisoning attacks, where there has been limited work in online settings. I agree with the observation that existing online attacks are easy to detect when there is a steady drop in model performance. The proposed strategy – of discretely preparing for a sudden and powerful attack – effectively exploits the online nature of the target. And the experiments show that it is surprisingly powerful.

The paper is generally well-executed. The attacks are defendable – they are derived by relaxing a formulation of the attacker’s objective as an optimization problem. However, I did find it difficult to follow the derivation in some places – e.g. seeing how eq (9) is obtained from eqs (7) and (8). I would like to see some of the missing steps/assumptions spelled out in an appendix. I appreciated the fact that online learning _and_ federated learning were both covered – effectively as two variations of the same framework.

One area in which the paper could be improved is in clarifying the threat model. I found that assumptions were mentioned in different places, making them difficult to piece together. My understanding is that a white box setting is assumed, where the attacker has full access to the model architecture, model parameters and the ability to poison all of the data received by the learning algorithm in each batch. In other words, a powerful attacker – although this is not out of the ordinary for data poisoning papers.

I found the experiments to be comprehensive, especially given the limited space. An ablation study on attack variants for the accumulation phase shows significant differences in the power of the attack. However to get a complete picture, I’d like to see the computational resources required for each variant. I was pleased to see an evaluation of the attack under an anomaly detection defense, however I think it’s difficult to judge the results in the absence of a benign reference. In particular, I’d like to see how the metrics compare with the values computed on unperturbed (benign) data.
Finally, I’d be interested to see an additional baseline to match the “vanilla” online poisoning referenced at the beginning of Section 3. It’s my understanding that the first row for each poisoned method in Table 1 is for a baseline where only a _single_ batch is poisoned. But what happens if an attacker continues to poison batches for the same number of iterations as the attacker running the accumulative attack? This would put the attackers on equal footing in terms of the horizon of the attack. The accumulative attack should be less powerful because its attempting to evade detection, but by how much?

Other comments:
- Line 108: Higher-order derivatives must apparently be computed for this attack, but how high? Are they capped at second-order?
- There are some unusual word choices in places e.g. retracement (L217)
- In various places, the model is said to be “irreparably broken” after an attack is launched. This sounds a little strange to me, as practical implementations of online learning would likely be able to revert to a previous checkpoint.
- For future work: it’d be interesting to develop intuition for how the model is affected by the accumulative phase of the attack. This may help in developing a defense for this attack.


**Time Spent Reviewing:**

4

---

> ### Author Response · Authors · 2021-08-10
> **Thank you for your supportive review**
>
> Thank you for appreciating our idea and approach, as well as the valuable suggestions. Below we address the detailed comments.
>
> ***Q1: About the derivations***
>
> Thank you for the suggestions We will include more derivation details in the revision. In fact, in the first line of Eq. (9), $\nabla_{\theta}\mathcal{L}(S_{t};\theta_{t})$ is the gradient on the clean batch $S_{t}$, and $\nabla_{\theta}\left(\nabla_{\theta}\mathcal{L}(S_{\text{val}},{\color{blue}\mathcal{A}}(\theta_{T}))^{\top}\nabla_{\theta}\mathcal{L}({\color{orange}\mathcal{P}}(S_{T});{\color{blue}\mathcal{A}}(\theta_{T}))\right)$ is the gradient of the minimization problem in Eq. (7). Solving the maximization problem in Eq. (9) is to make the accumulative gradient $\nabla_{\theta}\mathcal{L}({\color{blue}\mathcal{A}\_t}(S_{t});\theta_{t})$ to simultaneously align with $\nabla_{\theta}\mathcal{L}(S_{t};\theta_{t})$ and $\nabla_{\theta}\left(\nabla_{\theta}\mathcal{L}(S_{\text{val}},{\color{blue}\mathcal{A}}(\theta_{T}))^{\top}\nabla_{\theta}\mathcal{L}({\color{orange}\mathcal{P}}(S_{T});{\color{blue}\mathcal{A}}(\theta_{T}))\right)$, with a trade-off hyperparameter $\lambda$. In the second line of Eq. (9), since we cannot calculate ${\color{blue}\mathcal{A}}(\theta_{T})$ in advance during the accumulative phase, we greedily approximate ${\color{blue}\mathcal{A}}(\theta_{T})$ by $\theta_{t}$ in each accumulative step.
>
>
>
>
> ***Q2: About the threat model***
>
> Thanks for the suggestion. We'll further clarify the threat model in the final version. Indeed, we assume white-box accesses to model parameters and data points, in order to explore the effectiveness of the accumulative strategy. In the future work, many more advanced techniques (e.g., query-based, ensemble-based, transfer-based methods in the adversarial community) can be combined into our accumulative phase under different levels of black-box accesses.
>
>
> Note that although we assume white-box access to data points, we do not necessarily poison all the data received by the model. Namely, the ratios of poisoned data have different meanings in online/real-time and offline settings. In real-time settings, we only poison data during the accumulative phase. If we ask *the ratio of poisoned data points that are fed into the model*, the formula should be
> $$
> \frac{\textrm{Per-batch poisoning ratio}\times\textrm{Accumulative epochs}}{\textrm{Burin-in epochs}+\textrm{Accumulative epochs}}\textrm{.}
> $$
> So even if we use $100\\%$ per-batch poisoning ratio during the accumulative phase, the ratio of poisoned data points fed into the model is only $100\\% \times 2 / (40 + 2)\approx 4.76\\%$ in our settings. In contrast, if we poison $10\\%$ data in an offline dataset, then the expected ratio of poisoned data points fed into the model is also $10\\%$.
>
> Nevertheless, keeping a high poisoning ratio during the accumulative phase could still be challenging in practice. To this end, we constrain the poisoning operations to be imperceptible (e.g., less than $8/255$ under $\ell_\infty$-norm), while some previous works allow arbitrary modification on the poisoned data. Besides, our ablation studies in Table 2 also show that our method is still effective even if we use a $10\\%$ per-batch poisoning ratio.
>
>
>
>
> ***Q3: Computational resources and burden (and high-order derivative in Line 108)***
>
> As seen in Eq. (9), computing the high-order term requires executing the back-propagation twice. Fortunately, the objective for each back-propagation is a scalar (e.g., inner-product of two gradients), so the high-order term can be efficiently computed by reverse-mode automatic differentiation in modern packages like PyTorch. Besides, we only need to calculate the high-order term once out of the maximization loop of ${\color{blue}\mathcal{A}\_{t}}$, which largely saves computation. Empirically, we set the mini-batch as $100$, and use $10$-steps PGD attacks to execute poisoning. The running time for the vanilla poisoning attack is $2.33$ seconds per batch, and for our accumulative poisoning attack is $2.47$ seconds per batch, using a single RTX 1080-Ti GPU.
>
>
>
> ***Q4: Benign samples under detection***
>
> Thank you for the suggestion. We will include benign references on unperturbed data in the revision.
>
>
> ***Q5: Compare with multi-step poisoning attacks***
>
>
>
> As suggested, we compared with stronger baseline poisoning attacks, which can poison the model at multiple batches. Below we report the classification accuracy (\%) after the model is updated on the successive poisoned batches (i.e., without the accumulative phase). We apply different $\ell_{2}$-norm clip bounds to avoid arbitrarily large gradients fed into the federated model, using similar settings as in Fig. 1.
>
> > $\ell_{2}$-norm clip bound is $10$
>
> |  Poisoned batches | 1 |  2 | 3 | 4 | 5 |
> | :-----| :----: | :----: | :----: | :----: | :-----: |
> | Acc. after poison | 73.47 | 44.80 |14.96 | 10.08 |  10.09 |
>
> > $\ell_{2}$-norm clip bound is $1$
>
> |  Poisoned batches | 1 |  2 | 3 | 4 | 5 |
> | :-----| :----: | :----: | :----: | :----: | :-----: |
> | Acc. after poison | 83.38 | 83.62 | 83.72| 83.66 | 83.21 |
>
>
> In contrast, after using our accumulative phase, the model can be destructed to $27.66\\%$ / $61.87\\%$ under $\ell_{2}$-norm clipping of $10$ / $1$. These additional results again verify two advantages of our accumulative poisoning attacks: first is that our method can lead to more significant one-step destructive effects; second, our method can better bypass commonly used defense strategies like gradient clipping.
>
>
> ***Q6: Other comments***
>
> Thank you for the suggestions. We will polish our writing in the revision and make our claims more precise. In practice, even if the victim can revert the parameters to a previous state, this operation could lose important customer data from promotion days like Black Friday, or cause economic losses in high-frequency trading. Actually, it is an exciting topic to connect real-world scenarios with different attacking and defending strategies, which could be further explored in the future work. We will also explore and visualize how the model is affected by our accumulative phase.

---

> > ### Comment · Reviewer_C5BF · 2021-08-26
> > **RE: Thank you for your supportive review**
> >
> > Thanks for your responses.
> >
> > **Regarding Q1.**
> > Thanks, this explanation helps a lot. If my understanding is correct, it seems (9) is derived based on heuristic arguments - e.g. gradient alignment with two competing objectives. Please clarify this in the paper, as my initial impression was that (9) was a formal consequence of combining (7) and (8).
> >
> > **Regarding Q2.**
> > Thanks for agreeing to clarify the threat model. I think it's especially important to clarify how much data the attacker is perturbing. Which poisoning ratio are you reporting in Table 2? Is it the per-batch ratio or the cumulative ratio? If it's the cumulative ratio, then it appears as if some experiments (with $\mathcal{R} = 100%$) did not include burn-in epochs - i.e. the accumulative phase began immediately?
> >
> > **Regarding Q3.**
> > Thanks, I only point this out because the phrase "higher-order derivatives" is ambiguous. The term you're referring to in (9) looks second-order to me.
> >
> > **Regarding Q5.**
> > Thanks, it'd be great to include these results. This is a fairer comparison in my opinion, as both attackers are allowed to poison the same fraction of data points/updates. While the accumulative attack may not achieve the same damage, it should do a much better job at evading detection until the trigger is pulled.
> > One comment: the results in the second table are surprising to me. Why is there no significant drop in accuracy?

---

> > > ### Author Response · Authors · 2021-08-26
> > > **Thank you for the feedback**
> > >
> > > Thank you very much for the valuable feedback. Below we address the follow-up comments in detail.
> > >
> > > ***Q1: About Eq. (9)***
> > >
> > > Thank you for the suggestion. Actually, Eq. (9) is derived by the combination of minimizing the clean loss $\mathcal{L}(S\_{t};\theta\_{t})$ and minimizing the accumulative loss $\nabla\_{\theta}\mathcal{L}(S\_{\text{val}},{\color{blue}\mathcal{A}}(\theta\_{T}))^{\top}\nabla\_{\theta}\mathcal{L}({\color{orange}\mathcal{P}}(S\_{T});{\color{blue}\mathcal{A}}(\theta\_{T}))$ in Eq. (7), traded off by the hyperparameter $\lambda$. Eq. (8) indicates the update direction of parameters as $\nabla\_{\theta}\mathcal{L}({\color{blue}\mathcal{A}\_{t}}(S\_{t});\theta\_{t})$, while Eq. (9) aims to make this update direction to simultaneously minimize the clean loss and the accumulative loss. We will make this clearer in the revision.
> > >
> > >
> > > ***Q2: Poisoning ratio***
> > >
> > > All the poisoning ratios reported in our paper refer to the *per-batch ratio*, including those in Table 2. For examples, in Table 2, 100\% per-batch ratio corresponds to $\approx$4.76\% cumulative ratio; while 10\% per-batch ratio corresponds to $\approx$0.476\% cumulative ratio. All the experiments in our paper involve a burn-in phase, in which all the training samples are clean. We will make this clearer.
> > >
> > >
> > > ***Q3: About higher-order derivatives***
> > >
> > > Thank you for the suggestion. We will make our descriptions more precise in the revision.
> > >
> > >
> > > ***Q5: The results in the second table***
> > >
> > > There is no significant drop in accuracy because the vanilla poisoning attacks cannot bypass the gradient norm clip, e.g., $\ell\_{2}$-norm clip with bound of $1$. As seen in Table 3 of our paper, the one-step drop is not significant for the vanilla poisoning attacks. For the multi-step variant, the poisoning effect will be clipped out at each step, thus the accuracy drop is still limited. In contrast, our accumulative phase can facilitate bypassing the gradient clip operations. We will make this clearer.

---

### Official Review · Reviewer_94w7 · 2021-07-16

**Rating:** 6
**Confidence:** 3

**Summary:**

The paper proposes a poisoning attack for online/federated learning systems with the inclusion of an accumulation phase which then makes a trigger batch of poisons even more deleterious to the model.

**Limitations And Societal Impact:**

While the authors do briefly mention limitations in the appendix, it would be nice if they included a further discussion of this in the main body. They do mention that the attacker is assumed to have access to the model's parameters in the main body, but this isn't further discussed as a realistic or unrealistic assumption.

**Main Review:**

While I have experience with poisoning literature, I will add a disclaimer that I am not an expert at poisoning for online learning/federated learning.

### Strengths:
* The idea of an abrupt degradation of accuracy of an online model seems like an interesting and novel threat-model.
* The idea of "accumulating" the effects of poisons is also quite interesting.
* The drop in accuracy of the victim model is impressive.

### Weaknesses:
* The method should be explained more clearly and motivated in a better way. Heuristically, what's the "goal" of the accumulation poisons?
* It's not clear what capacity the attacker has. You mention that they have an l-infinity bound on the perturbations they are allowed to make, but are they allowed to perturb each client's update? If so this seems like a very strict assumption.
* To this end, if they are allowed to poison every update, a comparison to adaptations of other poisoning methods like the one found in Geiping et al. (cited in your work) would be in order.
* It would be nice if you described your online learning setup a bit further. You mention a "burn in" phase for the CIFAR-10 experiments involving 40 epochs of training. Could you describe how this differs from a standard supervised routine?


### Random Comments:
There are several grammatical errors that should be cleaned up.
For example:
* line 95: *becomes*
* line 115: *researches*

And a few others I forgot to mark down.

**Time Spent Reviewing:**

1

---

> ### Author Response · Authors · 2021-08-10
> **Thank you for your valuable review**
>
> Thank you for appreciating our idea and approach, as well as the valuable suggestions. Below we address the detailed comments.
>
> ***Q1: The goal of the accumulation poisons***
>
> In the accumulative phase, the attacker poisons the model in a stealthy way, such that the performance of the model is not affected, but it magnifies the destructive effect of the (poisoned) trigger batch. After the trigger batch is fed into the model, there will be a sudden drop of the model performance, before a monitor can perceive and intervene. Intuitively, the accumulative phase secretly accumulates a 'time bomb', which is later triggered by the trigger batch, as shown in Fig. 1. We'll make it clearer.
>
>
>
> ***Q2: The capacity of the attacker***
>
> First, we want to clarify that the ratios of poisoned data have different meanings in online/real-time and offline settings. Namely, in real-time settings, we only poison data during the accumulative phase. If we ask *the ratio of poisoned data points that are fed into the model*, the formula should be
> $$
> \frac{\textrm{Per-batch poisoning ratio}\times\textrm{Accumulative epochs}}{\textrm{Burin-in epochs}+\textrm{Accumulative epochs}}\textrm{.}
> $$
> So even if we use $100\\%$ per-batch poisoning ratio during the accumulative phase, the ratio of poisoned data points fed into the model is only $100\\% \times 2 / (40 + 2)\approx 4.76\\%$ in our settings. In contrast, if we poison $10\\%$ data in an offline dataset, then the expected ratio of poisoned data points fed into the model is also $10\\%$.
>
> Nevertheless, keeping a high poisoning ratio during the accumulative phase could still be challenging in practice. To this end, we constrain the poisoning operations to be imperceptible (e.g., less than $8/255$ under $\ell_\infty$-norm), while some previous works allow arbitrary modification on the poisoned data. Besides, our ablation studies in Table 2 also show that our method is still effective even if we use a $10\\%$ per-batch poisoning ratio.
>
> As to the case of federated learning, we propose a simple trick of recovered offset in Eq. (14), such that we can only manipulate one client to achieve any poisoned aggregated gradient. Namely, if we want to feed the model with a poisoned aggregated gradient ${\color{blue}\mathcal{A}}(G)$, and the aggregated clean gradients of other clients is $G'$, then we can manipulate a single client to contribute a gradient of ${\color{blue}\mathcal{A}}(G)-G'$, such that the total gradient is ${\color{blue}\mathcal{A}}(G)-G'+G'={\color{blue}\mathcal{A}}(G)$.
>
>
>
> ***Q3: Comparison with Geiping et al.***
>
> Thank you for the suggestion. However, as discussed in Section 2.1, Geiping et al. [1*] focus on backdoor attacks (targeted poisoning), while our work focuses on poisoning attacks (untargeted poisoning). Although we have different attacking goals, it would be interesting to modify the method in Geiping et al. to perform (untargeted) poisoning attacks, and compare/combine with our accumulative phase. We will further discuss this in the revision.
>
>
> ***Q4: Online learning setup***
>
> In our mimic experiments on CIFAR-10, the burn-in phase is the same as a standard supervised routine, which trains a model from scratch for $40$/$100$ epochs, using the SGD optimizer with momentum of $0.9$ and learning rate of $0.1$, the batch size is $100$, and the weight decay is $1\times 10^{-4}$.
>
>
>
>
> ***Q5: Random comments and societal impact***
>
> Thank you for pointing these out. We will further polish our writing and discuss the societal impact in the revision. As to the attacker assumptions, many more advanced techniques (e.g., query-based, ensemble-based, transfer-based methods in the adversarial community) can be combined into our accumulative phase under different levels of black-box accesses to the model parameters.
>
>
>
>
> **Reference**
>
>
> [1*] Geiping et al. Witches’ brew: Industrial scale data poisoning via gradient matching.

---

> > ### Comment · Reviewer_94w7 · 2021-08-19
> > **Response**
> >
> > Thanks to the authors for the detailed response. I have a few comments about it:
> >
> > > Q1: The goal of the accumulation poisons
> >
> > I appreciate the high-level intuition here. Although I was more wondering about specific/intuitive explanation of the optimization problem in Alg 1. The details there are a bit opaque.
> >
> > > Q2: The capacity of the attacker
> >
> > Thank you for clarifying this. However, this is a bit of a red flag for me. While you may only poison 4.76% of the data, it seems like you poison *all* of the data in the accumulative batches, which is an unrealistic assumption for poisoning federated learning systems - even if the poisons are imperceptible. Do you have ablations on this? What if you can only poison 10-20% during the accumulation phase? In my opinion, this is a significant limitation of this work, and at the very least needs to be discussed in more detail/prominence. However, I do appreciate the novel setting and threat model.
> >
> > > Q3: Comparison with Geiping et al.
> >
> > The gradient alignment objective found in Geiping et al. should be easily adaptable to whatever the poisoner's goals/setting may be. However, I acknowledge reimplementing another poisoning method in a rebuttal period is a lot to ask, so I'd just ask that you include comparisons to other approaches in a future updated version.
> >
> > > Q4/Q5:
> >
> > Thanks for the information/response on these!
> >
> > Overall, I'm still on the fence. I'm quite concerned as to the assumptions of the capacity of the attacker, but if these concerns are more fully addressed, I'm willing to recommend acceptance.

---

> > > ### Author Response · Authors · 2021-08-20
> > > **Thank you for the feedback**
> > >
> > > Thank you very much for the valuable feedback. Below we address the follow-up comments in detail. We hope you might view this as sufficient reason to further raise your score.
> > >
> > > ***Q1: The goal of the accumulation poisons***
> > >
> > > Thanks. We will make it clearer in the revision besides the high-level intuition. In fact, in Algorithm 1, $\nabla\_{\theta}\mathcal{L}(S\_{t}^{\nmid};\theta\_{t})$ is the detached gradient on the clean batch $S\_{t}$, which is the direction of *keeping accuracy*; while $G\_{t}=\nabla\_{\theta}\left(\nabla\_{\theta}\mathcal{L}(S\_{\text{val}},\theta\_{t})^{\top}\nabla\_{\theta}\mathcal{L}(S\_{T};\theta\_{t})\right)$ is the gradient of the minimization problem in Eq. (7), which is the direction of *maximizing destructive effect of the trigger batch $S\_{T}$*. Algorithm 1 iteratively updates ${\color{blue}\mathcal{A}\_{t}}$ and ${\color{orange}\mathcal{P}}$ by maximizing $H\_{t}$, which is  a trade-off between keeping accuracy and maximizing destructive effect of the trigger batch $S\_{T}$.
> > >
> > >
> > > ***Q2: The capacity of the attacker***
> > >
> > > Thank you for the suggestions. As suggested, we did ablation studies on the per-batch poisoning ratios in the federated learning cases. Empirically, we set loss scaling be $0.8$, and run the accumulative phase for $500$ steps (one epoch). We apply $\eta=10$ and $\ell\_{\infty}$ gradient clip with different poisoning ratios, and the results are shown below after the model is updated on the trigger batch:
> > >
> > > | Poisoning ratio (\%) | 80 |  60 | 40 | 20 | 10 |
> > > |:-----| :----: | :----: | :----: | :----: | :-----: |
> > > |Poisoned trigger (\%) | 16.77 | 37.87 | 52.85| 60.63 | 69.17 |
> > > | Accumulative phase + Clean trigger (\%) | 14.84 | 31.62 |45.11 | 52.01 |  63.76 |
> > >
> > > As we can see from the results, when the per-batch poisoning ratios are constrained, our accumulative strategy still consistently improves the effectiveness of the poisoning attacks. In the section of **More details on Q2**, we further explain different threat models of gradient clip for your reference. We'll include this discussion in the revision.
> > >
> > >
> > > ***Q3: Comparison with Geiping et al.***
> > >
> > > Thank you very much for the suggestion as well as the understanding of the difficulty in implementing a new method. As suggested, we will include the comparison with Geiping et al. in the updated revision, such that we have enough time to properly implement their method and provide a fair comparison.
> > >
> > >
> > >
> > >
> > >
> > >
> > >
> > > ***More details on Q2: The capacity of the attacker***
> > >
> > >
> > > In the federated learning cases, we denote $G$ as the aggregated gradient from the clients, and $G\_{n}$ as the gradient from the $n$-th client, where $n=1,\cdots,N$. We considered three threat models in our paper:
> > >
> > >
> > >
> > >
> > > > No gradient clip: $G=\sum\_{n}G\_{n}$
> > >
> > > In this case, we can only modify a *single* client, e.g., the $k$-th client, to achieve arbitrary poisoned aggregated gradient $G^{\textrm{poi}}$. Specifically, by the simple trick of recovered offset introduced in Eq. (14), we poison $G\_{j}$ to be ${\color{blue}\mathcal{A}}(G\_{j})$, where
> > > $$
> > > {\color{blue}\mathcal{A}}(G\_{j})=G^{\textrm{poi}}-\sum\_{n\neq j}G\_{n}\textrm{,}
> > > $$
> > > such that ${\color{blue}\mathcal{A}}(G\_{j})+\sum\_{n\neq j}G\_{n}=G^{\textrm{poi}}$. In our simulation experiments, we set batch size be $100$, and treat each data point as a client. So we can only poison the gradient on a single data point/client, i.e., the poisoning ratio is $1\\%$. Ideally, in this case, the poison ratio
> > > $$
> > > \frac{1}{\textrm{number of clients}}
> > > $$
> > > can be arbitrarily close to zero for a large number of clients. The empirical results correspond to the 'No clip' column in Table 3.
> > >
> > >
> > >
> > >
> > > > Gradient clip after aggregation: $G=\textbf{Clip}\_{\eta}(\sum\_{n}G\_{n})$
> > >
> > > Similar to the derivations above, in this case, we can still poison a *single* client, e.g., the $j$-th client, such that
> > > $$
> > > \textbf{Clip}\_{\eta}\left({\color{blue}\mathcal{A}}(G\_{j})+\sum\_{n\neq j}G\_{n}\right)=\textbf{Clip}\_{\eta}\left(G^{\textrm{poi}}\right)\textrm{,}
> > > $$
> > > where the poisoning ratio is still $1\\%$ in our simulation experiments, and can be arbitrarily close to zero as discussed above. The empirical results correspond to the '$\ell\_{2}$-norm clip bound' and '$\ell\_{\infty}$-norm clip bound' columns in Table 3.
> > >
> > >
> > >
> > >
> > >
> > > > Gradient clip before aggregation: $G=\sum\_{n}\textbf{Clip}\_{\eta}(G\_{n})$
> > >
> > > In this case, assuming that we can modify $M$ clients in the index set $S$, i.e., $|S|=M$ and the per-batch poison ratio is $\frac{M}{N}$. Specifically, for any $m\in S$, we poison $G\_{m}$ to be ${\color{blue}\mathcal{A}\_{m}}(G\_{m})$. Under the gradient clip constraint, we want to achieve $G^{\textrm{poi}}$ as well as we can, so we optimize the objective
> > > $$
> > > \min\_{{\color{blue}\mathcal{A}\_{m}}, m\in S}\left\\|G^{\textrm{poi}}-\left(\sum\_{m\in S}\textbf{Clip}\_{\eta}\left({\color{blue}\mathcal{A}\_{m}}(G\_{m})\right)+\sum\_{n\not\in S}\textbf{Clip}\_{\eta}\left(G\_{n}\right)\right)\right\\|\textrm{.}
> > > $$
> > > Under the mild condition that $\eta$ is small, the optimal solution of the above objective is $\forall m \in S$,
> > > $$
> > > {\color{blue}\mathcal{A}\_{m}}(G\_{m})=G^{\textrm{poi}}-\sum\_{n\not\in S}\textbf{Clip}\_{\eta}\left(G\_{n}\right)\textrm{.}
> > > $$
> > > and
> > > $$
> > > \sum\_{m\in S}\textbf{Clip}\_{\eta}\left({\color{blue}\mathcal{A}\_{m}}(G\_{m})\right)=\sum\_{m\in S}\textbf{Clip}\_{\eta}\left(G^{\textrm{poi}}-\sum\_{n\not\in S}\textbf{Clip}\_{\eta}\left(G\_{n}\right)\right)= \textbf{Clip}\_{{\color{red}M}\eta}\left(G^{\textrm{poi}}-\sum\_{n\not\in S}\textbf{Clip}\_{\eta}\left(G\_{n}\right)\right)\textrm{.}
> > > $$
> > > As we can see, poisoning more clients (i.e., larger $M$) can be regarded as relaxing the gradient clip constraint (i.e., relax from $\eta$ to $M\eta$). This case corresponds to the empirical results reported in the response of Q2.

---

> > > > ### Author Response · Authors · 2021-08-29
> > > > **Look forward to further feedback**
> > > >
> > > > Dear Reviewer 94w7,
> > > >
> > > > We thank you again for the timely feedback as well as the valuable comments. We hope you might find the response satisfactory and are looking forward to hearing from you about any further feedback.
> > > >
> > > > Best,
> > > > Authors

---

> > > > > ### Comment · Reviewer_94w7 · 2021-09-01
> > > > > **Response**
> > > > >
> > > > > Thank you for your thorough response and additional experiments. I have increased my score to reflect my concerns being addressed, and your commitment to addressing concerns. I would recommend that you add more discussion about the optimization problem in alg. 1 to improve readability, and I would also recommend that you make the capacity of the attacker in each setting abundantly clear, and include the table on varying attacker capacity in an updated version.

---

> > > > > > ### Author Response · Authors · 2021-09-02
> > > > > > **Thank you for the update**
> > > > > >
> > > > > > Thank you very much for the increase on rating and valuable feedback. We highly appreciate that. We'll try our best to further improve in the final version.

---

### Official Review · Reviewer_Gua5 · 2021-07-16

**Rating:** 7
**Confidence:** 4

**Summary:**

Data poisoning attacks usually target the offline setting and do not fully utilize the sequential nature of data utilization in the online learning scenario. This work aims to optimize the damage of a trigger batch during the training phase (akin to an availability attack) by adding selective perturbations to data leading up to the trigger batch. At that point, the trigger batch can cause significantly more damage than a vanilla trigger batch inserted in the data stream. The authors derive optimization requirements for the perturbations in this 'accumulative phase' and the trigger batch, and empirically demonstrate how the proposed attack can outperform other methods, all while being more robust to heuristics like gradient-clipping.

**Limitations And Societal Impact:**

### Limitations

1. As mentioned above, the authors should also consider a victim that keeps track of the best model based on validation loss (or even training loss) and see how effective the proposed attack can be in such a setting.

2. The lack of evaluation on any other datasets makes it hard to convince the reader of the effectiveness of the proposed attack.

---

### Ideas for future work

1. I would like to see in a future version how performance changes if the trigger batch is rejected by the victim at the batch at which it is inserted and how its effectiveness decreases with more data being fed into the model.

2. The method itself is pretty clever, but the victim can very quickly identify the trigger batch here. It would be exciting to see a variant of this method that can create a 'time-bomb' of sorts - causing a sudden drop in performance after some data batches. Hence, the victim would have a more challenging time identifying culprit data.


## Update

Most concerns raised have been addressed - changed rating accordingly

**Main Review:**

## Positive Feedback

1. The derivation of the proposed method is laid out quite well - with functional annotations (e.g., Equation 9) and color-coding (e.g., Equations 6,7) help follow the proposed algorithm's process.

2. The attack is robust enough against different hyper-parameters, like lambda values (Figure 3(b)). Performance stability is an excellent property to have and may suggest the proposed method being technically sound.

3. The benefit of the accumulative phase is most evident in the presence of gradient-clip operations. Performance drops are significantly better when an accumulative phase is included, further highlighting the proposed method's advantage. Although better performance in the standard scenario is a good sign, accumulative data to bypass gradient-clipping operations more effectively is valuable.

---

## Criticism

1. My main concern is with an adversary's power that can accumulate poisoned data, compared to one that inserts poisoned data at some given iteration. Although the former is better at utilizing the online nature of data being used by the model, it ultimately uses more poison points. Not only does this make it more likely for the attack to be detected, but it makes comparison with a simple slip-in baseline poisoning attack like an apples-to-oranges comparison. Although the proposed method objectively performs better, it is unclear how an adversary with a similar strength (perhaps insert poisoned data twice at specific intervals?) would hold against this. I understand that it is hard to quantify the adversary's strength in this case, but it is pretty clear that the adversary in the proposed work's threat model is more potent than one that only adds poison data once.

2. The inclusion of only one dataset makes empirical findings less convincing - it would be great if the authors could also include results for another dataset (could be even from the same domain, but a different task).

3. Assuming Whitebox access to random seeds feels like a powerful assumption for federated learning. If the adversary can access the random seeds, it can probably go a step further (in data manipulation or code control) or even adversarially pick seeds.

4. This work assumes that the victim relies on using the parameters at any given iteration as the primary model. The victim could have a basic validation set to spot sudden drops in performance and sound the alarm (while keeping track of the model with the best validation loss). Even more simply, it could keep track of the two most recent parameters, and reject data from batch $i + 1$ if it leads to a performance drop, and use parameters from state $i$. It would be nice if the authors can analyze the effect of skipping the trigger batch in such a setting.

---

## Minor comments

1. Equation 4: are the gradient updates scaled with $\frac{1}{N}$? If not, should the gradient update not be averaged?

2. Figure 2: What is the scale of values ere? For instance, how difference is 6.7 from 6.5 for metric A or 1500 v/s 1800 on Metric B. It would be great if the authors could include a  sentence or two to give the reader a better scope about how relevant values changes are for these metrics.

3. Figure 4 (and similar figures with image grids) are too small to notice any difference, even after zooming in. Please consider using a 5x5 (or lower) grid of images with a higher DPI.

4. This work seems quite similar to [1], in the sense of how both of them aim to exploit the sequential nature of data being fed in the training pipeline for a machine-learning model. Although this work is pretty recent, it would be worthwhile to talk about it (and perhaps in some future version add comparison experiments).
---

## References

[1] Shumailov, Ilia, et al. "Manipulating SGD with data ordering attacks." arXiv preprint arXiv:2104.09667 (2021).

**Time Spent Reviewing:**

3

---

> ### Author Response · Authors · 2021-08-10
> **Thank you for your supportive review**
>
> Thank you for appreciating our idea and approach, as well as the valuable suggestions. Below we address the detailed comments.
>
> ***Q1: Comparison with inserting poisoning data in some intervals***
>
> As suggested, we compared with stronger baseline poisoning attacks, which can poison the model at multiple batches. Below we report the classification accuracy (\%) after the model is updated on the successive poisoned batches (i.e., without the accumulative phase). We apply different $\ell_{2}$-norm clip bounds to avoid arbitrarily large gradients fed into the federated model, using similar settings as in Fig. 1.
>
> > $\ell_{2}$-norm clip bound is $10$
>
> |  Poisoned batches | 1 |  2 | 3 | 4 | 5 |
> | :-----| :----: | :----: | :----: | :----: | :-----: |
> | Acc. after poison | 73.47 | 44.80 |14.96 | 10.08 |  10.09 |
>
> > $\ell_{2}$-norm clip bound is $1$
>
> |  Poisoned batches | 1 |  2 | 3 | 4 | 5 |
> | :-----| :----: | :----: | :----: | :----: | :-----: |
> | Acc. after poison | 83.38 | 83.62 | 83.72| 83.66 | 83.21 |
>
>
> In contrast, after using our accumulative phase, the model can be destructed to $27.66\\%$ / $61.87\\%$ under $\ell_{2}$-norm clipping of $10$ / $1$. These additional results again verify two advantages of our accumulative poisoning attacks: 1) our method can lead to more significant one-step destructive effects; and 2) our method can better bypass the commonly used defense strategies like gradient clipping.
>
>
>
>
> ***Q2: Results on other datasets***
>
> As suggested, we also mimicked the real-time data training using the MNIST dataset, on which the attacking is more challenging to succeed and defending is easier. We employed the SGD optimizer with momentum of $0.9$ and weight decay of $1\times 10^{-4}$. The initial learning rate is $0.1$, and the mini-batch size is $100$. In the burn-in phase, we pre-trained for $10$ epochs to achieve a classification accuracy of $99\\%$. Below, we report classification accuracy (\%) in the online learning and federated learning cases on MNIST.
>
>
> > Online learning cases on MNIST with burn-in phase be $10$ epochs
>
> | Method | Acc. before trigger | Acc. After trigger | $\Delta$ |
> | :-----| :----: | :----: | :----: |
> | Poison trigger | 98.29 | 97.99 | 0.30|
> | Accumulative phase <br> + Poison trigger |  96.83$\pm$0.18 | 85.78 $\pm$+0.40 | 11.05 $\pm$ 0.22|
>
>
> > Federated learning cases on MNIST with burn-in phase be $10$ epochs
>
> | Method | Loss scaling |  No clip | bound 10 | bound 1 | bound 0.1|
> | :-----| :----: | :----: | :----: | :----: | :-----: |
> | Poison trigger | 1 |  98.27 | 98.27 | 98.27 | 98.28|
> | Poison trigger | 10 |  95.49 | 95.49 | 98.12 |98.28|
> | Poison trigger | 20 |  84.09 |  89.24 | 98.12 |  98.28 |
> | Poison trigger | 50 |  31.93 | 89.24 | 98.12 | 98.28 |
> | Accumulative phase | 0.08 | 22.49  | 22.49 | 32.87| 51.28|
>
>
>
> The accumulative phase runs for $200$ steps ($\frac{1}{3}$ epoch). In the online learning cases, we craft accumulative poisoning examples by $10$-steps PGD with $\epsilon=16/255$, and the step size is $\alpha=2/255$. These additional results demonstrate that our method can still work well on simple datasets like MNIST.
>
>
>
> ***Q3: White-box access***
>
> Thanks for the suggestion. Indeed, white-box accesses are powerful assumptions, where many other techniques like code/seeds control can be applied. Nevertheless, in this paper we do not intend to design a *strongest* poisoning attack under white-box assumptions; instead, we propose the accumulative strategy as a new module to amplify the one-step destructive effect. Many more advanced techniques (e.g., query-based, ensemble-based, transfer-based methods in the adversarial community) can be combined into our accumulative phase under different levels of black-box accesses.
>
>
>
>
> ***Q4: Effects of skipping trigger***
>
> Even if the victim rejects the trigger batch $T+1$, the model parameters from state $T$ are already poisoned by the accumulative phase, and may be triggered again in the following updates. To make sure the model is well-behaved and not poisoned, practical recommendation/financial systems may choose to reset the parameters to very early state, e.g., state $0$. In real-world cases, this reset operation could lose important customer data from promotion days like Black Friday, or cause economic losses in high-frequency trading. Actually, it is an interesting topic to connect real-world scenarios with different attacking and defending strategies, which could be further explored in the future work.
>
>
>
>
>
> ***Q5: Minor comments***
>
> Thank you for pointing these out. In Eq. (4), the gradient updates should be scaled by $\frac{1}{|I_{t}|}$, where $|I_{t}|$ is the number of clients in $I_{t}$. In Fig. 2, the metric value for KD is the kernel density, for LID is the negative local dimension, for GDA and GMM is the likelihood. We will modify these parts and more clearly explain the settings. As to Fig. 4, we will include pictures with better DPI.
>
> The work [1*] is quite intriguing, which manipulates the order of data feeding to poison the model. It would be an interesting future work to combine [1*] with our method, e.g., performing the accumulative phase by manipulating the order of data feeding, rather than adversarial perturbations. We will further discuss this in the revision.
>
>
>
>
> ***Q6: Ideas for future work***
>
> Thank you for the interesting ideas. We would like to further improve our methods to be robust against different real-world scenarios.
>
>
> **Reference**
>
>
> [1*] Shumailov, Ilia, et al. "Manipulating SGD with data ordering attacks." arXiv preprint arXiv:2104.09667 (2021).

---

> > ### Comment · Reviewer_Gua5 · 2021-08-26
> > **Rebuttal Acknowledgement**
> >
> > I would like to thank the reviewers for their detailed responses. Most of my concerns have been addressed (and I have updated my ratings accordingly), but the threat model itself is not very convincing to me.

---

> > > ### Author Response · Authors · 2021-08-27
> > > **Thank you again**
> > >
> > > Thank you again for the valuable comments. As to the threat model, we will involve evaluations in the revision under more limited accesses (e.g., without knowing the random seeds). Proposing stronger accumulative variants under black-box settings is also an essential direction of our future work.

---

### Official Review · Reviewer_cbZK · 2021-07-17

**Rating:** 6
**Confidence:** 4

**Summary:**

This paper presents a novel poisoning attack strategy for online learning settings, where the parameters of the machine learning model are updated as new batches of data are collected. For this, the authors propose an accumulative poisoning attack strategy with two phases. In the first phase, the attacker poisons the model in a stealthy way, so that the performance of the model is not affected, so that it magnifies the destructive effect of the poisoning attack conducted in the second phase, where the attacker injects poisoning points in a batch of samples to decrease the performance of the model. The authors shows that this attack strategy can also be applied for compromising federated learning.

**Ethical Concerns:**

No ethical concerns.

**Limitations And Societal Impact:**

No comments.

**Main Review:**

The idea and the algorithm proposed in the paper is very interesting and, to my best knowledge, novel. I really like the idea of having a preliminary phase where the attacker, in a stealthy way, manipulates the learning algorithm without affecting the performance, so that the impact of the attack is maximized in the second phase. In this sense, I think that the differences with respect to backdoor attacks are clear and adequately justified in Section 3.4.

The presentation and justification of the method are fine. However, following some of the steps in the derivation of the attack are not straightforward (see for example the derivation of (9)) and I think that more details about the steps (in the appendix) are needed to help readers to understand and follow better the derivation of the attack strategy. With respect to the attack formulation, I miss the analysis with the computational complexity of the proposed attack strategy.

My main concerns about the paper are on the experimental evaluation:

(1) The authors just considered a single dataset, CIFAR, for the experimental evaluation. In this context, CIFAR is an easier dataset to compromise, given that the dataset is quite multi-modal with low-resolution images and the accuracy of the baseline model is less than 85%, which makes it more unstable against data poisoning attacks. It would be interesting to analyse the performance of this attack strategy in other benchmarks. For example, in MNIST, where poisoning with stealthy strategies can be more difficult to achieve.

(2) The assumptions about the attacker are very strong in most of the experiments. In most cases, except for Table 2, the authors assume that the attacker is in control of 100% of the training points (please, correct me if I am wrong), which is quite unrealistic in most cases and provides a misleading view of the effectiveness of the attack. In the case of federated learning, it is unclear what is the fraction of malicious clients present in the platform. From the previous experiments I assume that is 100% of the clients, which explains the results reported in Tables 3 and 4.

(3) Defensive techniques are not properly analyzed. For example, Figure 2 just report the metric values (which ones?) for the different anomaly detectors to show that the data points generated during the accumulative phase are stealthier. But is the performance of the model affected in the presence of such anomaly detectors and what is their effectiveness to detect the attack points generated using the vanilla and the proposed strategy (for example by looking at the AUC, false positives/negatives or the F1 score of the detector). Similarly, it would be interesting to analyze to test the attack against robust optimization methods such as Sever (Diakonikolas et al. “Sever: A Robust Meta-Algorithm for Stochastic Optimization”) or Rubinstein et al. “Antidote: Understanding and Defending against Poisoning of Anomaly Detectors” by, for example, performing a transfer attack. Similarly, in federated learning settings, it would be interesting to test the performance of the model against robust aggregation methods such as Krum (Blanchard et al. “Machine learning with adversaries: Byzantine tolerant gradient descent”) or median-based robust aggregation techniques (Yin et al. “Byzantine-Robust Distributed Learning: Towards Optimal Statistical Rates”), just to cite some.

Side comment: The pictures in Figure 4 are too small to appreciate differences. It would also be good to show some pictures of poisoning points generated with vanilla approaches to highlight the differences between attacks.


**Time Spent Reviewing:**

5.5

---

> ### Author Response · Authors · 2021-08-10
> **Thank you for your valuable review**
>
> Thank you for appreciating our idea and approach, as well as the valuable comments. Below we address the detailed comments.
>
> ***Q1: About the derivations***
>
> Thank you for the suggestions. We will include more derivation details in the revision. In the first line of Eq. (9), $\nabla_{\theta}\mathcal{L}(S_{t};\theta_{t})$ is the gradient on the clean batch $S_{t}$, and $\nabla_{\theta}\left(\nabla_{\theta}\mathcal{L}(S_{\text{val}},{\color{blue}\mathcal{A}}(\theta_{T}))^{\top}\nabla_{\theta}\mathcal{L}({\color{orange}\mathcal{P}}(S_{T});{\color{blue}\mathcal{A}}(\theta_{T}))\right)$ is the gradient of the minimization problem in Eq. (7). Solving the maximization problem in Eq. (9) is to make the accumulative gradient $\nabla_{\theta}\mathcal{L}({\color{blue}\mathcal{A}\_t}(S_{t});\theta_{t})$ to simultaneously align with $\nabla_{\theta}\mathcal{L}(S_{t};\theta_{t})$ and $\nabla_{\theta}\left(\nabla_{\theta}\mathcal{L}(S_{\text{val}},{\color{blue}\mathcal{A}}(\theta_{T}))^{\top}\nabla_{\theta}\mathcal{L}({\color{orange}\mathcal{P}}(S_{T});{\color{blue}\mathcal{A}}(\theta_{T}))\right)$, with a trade-off hyperparameter $\lambda$. In the second line of Eq. (9), since we cannot calculate ${\color{blue}\mathcal{A}}(\theta_{T})$ in advance during the accumulative phase, we greedily approximate ${\color{blue}\mathcal{A}}(\theta_{T})$ by $\theta_{t}$ in each accumulative step.
>
>
> ***Q2: Computational complexity***
>
>
> As seen in Eq. (9), the high-order term only appears once out of the maximization loop of ${\color{blue}\mathcal{A}\_{t}}$, and can be efficiently computed by reverse-mode automatic differentiation in modern package like PyTorch. Empirically, we set the mini-batch as $100$, and use $10$-steps PGD attacks to execute poisoning. The running time for the vanilla poisoning attack is  $2.33$ seconds per batch, and for our accumulative poisoning attack is $2.47$ seconds per batch, using a single RTX 1080-Ti GPU. We'll make this more explicit.
>
>
> ***Q3: Experiments under stronger baseline models***
>
>
> Thank you for your suggestion. We experimented in the case of a stronger baseline/burn-in model, which is pre-trained on clean data for $100$ epochs (compared to $40$ epochs in, e.g., Fig. 1) on CIFAR-10, and achieves $89.92\\%$ accuracy. We applied ResNet18 as the model architecture, and employed the SGD optimizer with momentum of $0.9$ and weight decay of $1\times10^{-4}$. The initial learning rate is $0.1$, and the mini-batch size is 100. The classification accuracy (\%) in the online learning and federated learning cases are reported below:
>
>
>
> > Online learning cases on CIFAR-10 with burn-in phase be $100$ epochs
>
> | Method | Acc. before trigger | Acc. After trigger | $\Delta$ |
> | :-----| :----: | :----: | :----: |
> | Poison trigger | 89.92 | 86.86 | 3.06|
> | Accumulative phase <br> + Poison trigger |  87.64$\pm$0.22 |76.89 $\pm$+0.39 | 10.75 $\pm$ 0.25|
>
>
> > Federated learning cases on CIFAR-10 with burn-in phase be $100$ epochs
>
> | Method | Loss scaling |  No clip | bound 10 | bound 1 | bound 0.1|
> | :-----| :----: | :----: | :----: | :----: | :-----: |
> | Poison trigger | 1 |  89.34 | 89.34 | 89.91 | 89.99 |
> | Poison trigger | 10 |  45.29 | 84.45 | 89.91 | 89.99 |
> | Poison trigger | 20 |  16.62 | 84.45 | 89.91 | 89.99 |
> | Poison trigger | 50 |  10.24 | 84.45 | 89.91 | 89.99 |
> | Accumulative phase | 0.01 | 80.35 | 80.35 |  80.35 | 83.32|
> | Accumulative phase | 0.02 |  25.45 | 25.45 | 25.45 | 76.06 |
> | Accumulative phase | 0.05 | 12.03 | 12.03|  15.53 | 70.00|
> | Accumulative phase | 0.08 | 11.07  |11.07   |14.23  | 64.74|
>
>
>
> The accumulative phase runs for $200$ steps ($0.4$ epoch). In the online learning cases, we craft accumulative poisoning examples by $10$-steps PGD with $\epsilon=16/255$, and the step size is $\alpha=2/255$. These additional results demonstrate the consistent effectiveness of our method even against stronger baseline models.
>
>
>
>
>
>
> ***Q4: Experiments on other datasets***
>
> As suggested, we also mimicked the real-time data training using the MNIST dataset. We employed the SGD optimizer with momentum of $0.9$ and weight decay of $1\times 10^{-4}$. The initial learning rate is $0.1$, and the mini-batch size is $100$. In the burn-in phase, we pre-trained for $10$ epochs to achieve a classification accuracy of $99\\%$. Below, we report classification accuracy (\%) in the online learning and federated learning cases on MNIST.
>
>
> > Online learning cases on MNIST with burn-in phase be $10$ epochs
>
> | Method | Acc. before trigger | Acc. After trigger | $\Delta$ |
> | :-----| :----: | :----: | :----: |
> | Poison trigger | 98.29 | 97.99 | 0.30|
> | Accumulative phase <br> + Poison trigger |  96.83$\pm$0.18 | 85.78 $\pm$+0.40 | 11.05 $\pm$ 0.22|
>
>
> > Federated learning cases on MNIST with burn-in phase be $10$ epochs
>
> | Method | Loss scaling |  No clip | bound 10 | bound 1 | bound 0.1|
> | :-----| :----: | :----: | :----: | :----: | :-----: |
> | Poison trigger | 1 |  98.27 | 98.27 | 98.27 | 98.28|
> | Poison trigger | 10 |  95.49 | 95.49 | 98.12 |98.28|
> | Poison trigger | 20 |  84.09 |  89.24 | 98.12 |  98.28 |
> | Poison trigger | 50 |  31.93 | 89.24 | 98.12 | 98.28 |
> | Accumulative phase | 0.08 | 22.49  | 22.49 | 32.87| 51.28|
>
>
>
> The accumulative phase runs for $200$ steps ($\frac{1}{3}$ epoch). In the online learning cases, we craft accumulative poisoning examples by $10$-steps PGD with $\epsilon=16/255$, and the step size is $\alpha=2/255$. These additional results demonstrate that our method can still work well on simple datasets like MNIST, on which the attacking is more challenging to succeed.
>
>
>
>
>
>
> ***Q5: Poisoning ratios***
>
> First, we want to clarify that the ratios of poisoned data have different meanings in online/real-time and offline settings. Namely, in real-time settings, we only poison data during the accumulative phase. If we ask *the ratio of poisoned data points that are fed into the model*, the formula should be
> $$
> \frac{\textrm{Per-batch poisoning ratio}\times\textrm{Accumulative epochs}}{\textrm{Burin-in epochs}+\textrm{Accumulative epochs}}\textrm{.}
> $$
> So even if we use $100\\%$ per-batch poisoning ratio during the accumulative phase, the ratio of poisoned data points fed into the model is only $100\\% \times 2 / (40 + 2)\approx 4.76\\%$ in our settings. In contrast, if we poison $10\\%$ data in an offline dataset, then the expected ratio of poisoned data points fed into the model is also $10\\%$.
>
> Nevertheless, keeping a high poisoning ratio during the accumulative phase could still be challenging in practice. To this end, we constrain the poisoning operations to be imperceptible (e.g., less than $8/255$ under $\ell_\infty$-norm), while some previous works allow arbitrary modification on the poisoned data. Besides, our ablation studies in Table 2 also show that our method is still effective even if we use a $10\\%$ per-batch poisoning ratio.
>
> As to the case of federated learning, we propose a simple trick of recovered offset in Eq. (14), such that we can only manipulate one client to achieve any poisoned aggregated gradient. Namely, if we want to feed the model with a poisoned aggregated gradient ${\color{blue}\mathcal{A}}(G)$, and the aggregated clean gradients of other clients is $G'$, then we can manipulate a single client to contribute a gradient of ${\color{blue}\mathcal{A}}(G)-G'$, such that the total gradient is ${\color{blue}\mathcal{A}}(G)-G'+G'={\color{blue}\mathcal{A}}(G)$.
>
> We'll make this clearer in the final version.
>
>
> ***Q6: Defense techniques***
>
> In Fig. 2, the metric value for KD is the kernel density, for LID is the negative local dimension, for GDA and GMM is the likelihood. We will more clearly explain the settings in the revision. The performance of the model will not be affected by KD/LID/GDA/GMM, since they are fitted in the learned feature space after the model is trained. While as shown in Fig. 3(a), gradient clipping as a defense will affect the model performance. In the revision, we will report the ROC-AUC scores under detection defenses.
>
>
> Thank you for the suggestions on evaluating under more interesting defenses [1*,2*,3*,4*]. We will investigate these works and try our best to provide comprehensive results in the revision. More explorations would be our essential future work.
>
>
> ***Q7: Side comment***
>
> Thank you for the suggestion. We will make Figure 4 clearer and provide pictures poisoned by vanilla approaches in the revision.
>
>
>
>
> **Reference**
>
> [1*] Diakonikolas et al. Sever: A Robust Meta-Algorithm for Stochastic Optimization.
>
> [2*] Rubinstein et al. Antidote: Understanding and Defending against Poisoning of Anomaly Detectors.
>
> [3*] Blanchard et al. Machine learning with adversaries: Byzantine tolerant gradient descent.
>
> [4*] Yin et al. Byzantine-Robust Distributed Learning: Towards Optimal Statistical Rates.

---

> > ### Author Response · Authors · 2021-08-28
> > **Look forward to further feedback**
> >
> > Dear Reviewer cbZK,
> >
> > We thank you again for the valuable comments. We hope you might find the response satisfactory and are looking forward to hearing from you about any further feedback.
> >
> > Best,
> > Authors

---

> > > ### Comment · Reviewer_cbZK · 2021-08-29
> > > **Response to the authors**
> > >
> > > Thank you very much for your detailed reponse to the different points in my review and thank you very much for your effort in running further experiments to address some of my concerns. Although I still think that the threat model is not very clear, I think that the idea explored in the paper is very original. With the results that the authors have provided in their response, which support better the claims made in the paper, even if I think that a more comprehensive set of defenses should be explored, I am increasing my score.

---

> > > > ### Author Response · Authors · 2021-08-30
> > > > **Thanks for the update**
> > > >
> > > > Thank you very much for the increase on rating and valuable feedback. We highly appreciate that. We'll try our best to further improve in the final version.

---

### Author Response · Authors · 2021-08-19
**Looking forward to further feedbacks**

Dear Reviewers,

Thank you again for your valuable comments and suggestions, which are really helpful for us. We have posted responses to the detailed concerns.

We totally understand that this is a quite busy period, since the reviewers may be responding to the rebuttal of other assigned papers or rushing for deadlines.

We deeply appreciate it if you can take some time to return further feedback on whether our responses solve your concerns. If there are any other comments, we will try our best to address them.

Best,

The authors

---

### Decision · Program_Chairs · 2021-09-27

**Decision:**

Accept (Poster)

**Comment:**

The reviewers were overall positive regarding the paper, and they were satisfied by the rebuttal. After the discussion period, the reviewers requested for the following things to be included in the paper:

1. More description of the optimization problem in Algorithm 1.
2. A  table about the capacity of the attacker in the federated learning setting.
3. Include the additional empirical evaluations mentioned mentioned by the authors during the rebuttal period.